# BlockAuth: A blockchain-based framework for secure vehicle authentication and authorization

**Gauhar Ali** [iD] *, **Mohammed ElAffendi, Naveed Ahmad**

EIAS Data Science and Blockchain Lab, College of Computer and Information Sciences, Prince Sultan University, Riyadh, Saudi Arabia

* gali@psu.edu.sa

**Data Availability Statement:** The implementation code and sample access control policies are available on GitHub at https://github.com/gali675/BlockAuth.git.

## Abstract

Intelligent Transport System (ITS) offers inter-vehicle communication, safe driving, road condition updates, and intelligent traffic management. This research intends to propose a novel decentralized "BlockAuth" architecture for vehicles, authentication, and authorization, traveling across the border. It is required because the existing architects rely on a single Trusted Authority (TA) for issuing certifications, which can jeopardize privacy and system integrity. Similarly, the centralized TA, if failed, can cause the whole system to collapse. Furthermore, a unique "Proof of Authenticity and Integrity" process is proposed, redirecting drivers/vehicles to their home country for authentication, ensuring the security of their credentials. Implemented with Hyperledger Fabric, BlockAuth ensures secure vehicle authentication and authorization with minimal computational overhead, under 2%. Furthermore, it opens up global access, enforces the principles of separation of duty and least privilege, and reinforces resilience via decentralization and automation.

## 1. Introduction

Intelligent Transport System (ITS) [1] has been a forefront research area in recent years. ITS makes vehicle transportation safe and easy by making vehicles intelligent. ITS communication architecture consists of Intelligent Transport System-Stations (ITS-Ss), i.e., Road Side Units (RSUs), and vehicles. Every vehicle has an On-Board Unit (OBU) for communication with other vehicles. Moreover, ITS has defined two means of communication [2] i.e., Vehicle-to-Vehicle (V2V) and Vehicle-to-Infrastructure (V2I). Both V2V and V2I communications are defined by IEEE 802.11P standard [3]. Each ITS-S broadcasts Cooperative Awareness Messages (CAMs) [4] periodically to their neighbors. The vehicles and RSUs broadcast CAMs to attain road safety. CAMs are used for emergency vehicle warnings, emergency brakes, etc. Similarly, Decentralized Environmental Notification Message (DENMs) [5] contains information like traffic congestion or a road hazard. Unlike CAMs, DENMs are triggered in response to an event. It broadcasts to warn drivers/vehicles about hazardous road events.

The vehicles and RSUs must be registered with the Certification Authority. The Enrollment Certification Authority (ECA) [6] generates two types of certificates for data signing and data

**Funding:** The author(s) received no specific funding for this work.

**Competing interests:** The authors have declared that no competing interests exist.

**Abbreviations:** BC, Blockchain; BSM, Blockchain Service Manager; CA, Certification Authority; CAMs, Cooperative Awareness Messages; DENMs, Decentralized Environmental Notification Messages; ECA, Enrollment Certification Authority; ITS, Intelligent Transportation System; ITS-Ss, Intelligent Transport System-Stations; LTC, Long Terms Certificates; $o_i$, resource or service; $p_j$, Permission on the resource or service; PKI, Public Key Infrastructure; PoAI, Proof of Authenticity and Integrity; RCA, Regional Certification Authority; RSU, Road Side Unit; $s_i$, user or vehicle or RSU; SP, Service Provider; TA, Trusted Authority; TCA, Transition Certification Authority; V2I, Vehicle-to-Infrastructure; V2V, Vehicle-to-Vehicle.

encryption. Moreover, the Transition Certification Authority (TCA) generates the transition certificates for vehicles and RSUs to invoke transactions on the Blockchain (BC).

At present, ITS has a number of challenges, i.e., integrity, privacy, centralization, and trust among ITS network nodes [7, 8]. In [9], the authors have introduced a trusted entity called Delegation Service (DS). The DS is responsible for cross-domain access control. It redirects the vehicle to its parent domain CA for authentication. After successful authentication, DS authorized the vehicle for a service based on access control policies attached to the service. Similarly, in [10, 11], the authors have suggested a novel access control framework for virtual coalitions. In the virtual coalition, there is a lack of trust among the member organizations. Therefore, the authors have introduced a centralized trusted mediator. It performs authentication and privacy preservation for clients as well as service owners. In [9–11], a centralized trusted entity is used to implement access control mechanisms. However, the centralized entity, if failed, could crumble access control mechanisms. Moreover, the trusted entity has low resilience to different attacks or hacks like Denial of Service (DOS) attacks, Dynamic Denial of Service (DDOS) attacks, and Sybil attacks [12]. In the centralized system, participants authorize the trusted entity to make authorization decisions. Hence, the trusted entity can alter service provider policies to allow illegal authorization. Also, the principles of the separation of duty and least privilege are not considered during the implementation of the authorization service. Moreover, these authentication and authorization mechanisms require high computation power and have complex implementations. Therefore, these mechanisms are not suitable for ITS.

Similarly, Public Key Infrastructure (PKI) is used for vehicle and RSU authentication and authorization in the literature [13–15]. However, like centralized authentication mechanisms, PKI has the same limitations. Moreover, PKI does not have sensing functionalities like vehicles. Therefore, it does not have enough information about the reality of the situation.

## 1.1 Challenges

The following are the problems within the existing literature.

- A centralized trusted access control service is a single point of failure and has low resilience to different attacks.

- As a result of inadequate implementation of the separation of duty and least privilege principle, the users are granted excessive privileges.

- This trusted third-party service can perform illegal authorization by altering stored authorization policies

- This trusted service can expose user credentials without user consent.

To address these problems, we propose a BC-based architecture for cross-border vehicle authentication and authorization called BlockAuth. Let's suppose a scenario where different countries form a virtual coalition to facilitate vehicle traveling across the border. Each country in the coalition has a local CA to manage local/internal vehicle certificates. Likewise, the global CAs originate from various countries and collaboratively constitute a virtual coalition known as the BC network. The BC has a single smart contract. The BC stores access control policies for global/external vehicles. During the authentication of an external vehicle, the smart contract broadcasts the vehicle "join" request to all the local CAs. Subsequently, the CA responsible for the initial vehicle registration, authenticates and provides the Proof of Authenticity and Integrity (PoAI). Similarly, during authorization, the smart contract authorizes the vehicle based on SP access control policies. We use BC to store and validate access control policies.

### 1.2 Our contributions

The following are the main contributions of this research study.

- The single trusted authentication and authorization service is substituted with BC because it can collapse the entire system and expose it to several attacks.

- The authorization procedure is strengthened with the implementation of the separation of duty and least privilege principle in the smart contract.

- Our proposed BlockAuth architecture stores authorization policies on the immutable ledger of the BC, effectively preventing any illegal authorization.

- The proposed BlockAuth architecture allows drivers/vehicles to get authentication from their parent countries, thus preventing credentials disclosure.

### 1.3 Significance of the study

Collectively, the highlighted contributions hold immense practical implications. The replacement of the centralized access control service with BC technology mitigates the vulnerability that could potentially lead to the collapse of the entire system and susceptibility to various attacks. Similarly, storing authorization policies on the BC not only ensures that illegal authorizations are effectively prevented but also enhances the security and integrity of the system. Additionally, enabling authentication from parent countries ensures the privacy of credentials, effectively averting the risks associated with credentials disclosure. Thus, this research significantly advances the realm of authentication and authorization, offering enhanced security and efficiency for modern systems.

The organization of the research paper is described as follows. In Section II, we perform a lit-erature review and debate the related works. The core components, system operations, and formal modeling of the BlockAuth framework are presented in section III. BlockAuth framework implementation, experimental results, and security analysis are debated in Section IV. Lastly, we draw conclusions in Section V.

## 2. Preliminaries and related works

In this section, we discuss BC, components of the BC, working of the BC, and related works on the BC. The.

### 2.1 Blockchain

Blockchain [16] is a mesh network of untrusted nodes. A consensus mechanism is used to develop trust among BC nodes. The miners group different transactions into blocks and validate these transactions using a consensus mechanism. Therefore, after validation, the blocks are attached to a chain of blocks [17]. The following are the core components of BC.

- Distributed and Immutable ledger: BC nodes maintain a copy of immutable ledger [18, 19]. These distributed ledgers are synchronized using a replication mechanism. It allows the BC to work if some of the nodes are malfunctioned.

- Asymmetric cryptography: Asymmetric cryptography consists of two types of keys, i.e., public and private keys. Therefore, BC uses a public/private key for signing and encryption/decryption of transactions to ensure integrity and authentication [20].

- Consensus Mechanism: BC nodes execute a consensus mechanism [21] to add a block to the chain. It allows nodes to agree on the current state of the BC network.

- Peer-to-peer (P2P) Network: A P2P network is a mesh network of computers. Moreover, the BC network uses a consensus algorithm to develop consensus for adding a block to the chain.

Additionally, BC technology has revolutionized numerous industries by offering decentralization, transparency, and security in various procedures. Therefore, the applications of BC technology spread across numerous fields i.e., Healthcare [22, 23], Cyber-physical Systems [24], Supply Chain Management [25], Internet of Things [26, 27], etc.

**2.1.1 How BC works.** BC builds a chain of blocks. These blocks are linked together using block hashes such that each block contains a hash of the previous block [28]. The hash preserves the integrity of the stored blocks and stops them from being altered [29]. Moreover, a block includes transactions, Proof of Work, and a header. Also, each transaction is time-stamped.

BC is a P2P network of interconnected nodes. The nodes with greater computing power are called miners. Every miner fetches a group of a transaction from the transaction pool and forms a block. However, only one miner will add his block to the chain and get the reward. Therefore, miners try to solve a mathematical puzzle called PoW [30]. A miner can add its block to the chain if it finds PoW first. Then, it broadcasts the block with PoW to all the other miners for validation. Every miner validates the PoW and adds the block to his local ledger if the validation is successful.

The consensus mechanism allows BC peers to agree on a single value [31]. The PoW, a complex mathematical puzzle, is used as a consensus mechanism in Bitcoin. The miner who solves the puzzle is rewarded. In Bitcoin, a new block of a transaction is added to the BC every 10 minutes approximately [32]. Unlike PoW, a validator is selected by an election process in PoS. A validator having a large amount of money, i.e., a stack, has a greater chance of being selected as a validator for the next block. Unlike PoW, PoS requires less computational power. Similarly, the Byzantine Fault Tolerant (BFT) consensus mechanism ensures that every non-malicious peer has the same BC state. The BFT consensus mechanism has the ability to reach enough consensus despite the fact that some of the BC peers are malicious. Hyperledger Fabric has implemented the BFT consensus mechanism.

**2.1.2 Smart contract.** The smart contract is an application that contains pre-defined rules to control communication among entities [33]. Nick Szabo was the first who propose the smart contract. It runs automatically within a BC when a particular criterion is met. It is used to transfer any resource having value without the need for a trusted mediator, i.e., a bank. Ethereum and Hyperledger are well-known platforms that have implemented smart contracts.

**2.1.3 Implementations of smart contract in Blockchain.** In this section, we discuss the well-known implementations of BC i.e., Ethereum and Hyperledger.

- Ethereum: An Ethereum [34] BC is a decentralized ledger of transaction. Ethereum BC runs on top of Ethereum Virtual Machine (EVM). In Ethereum, a smart contract is written in a programming language like Solidity [35]. Before execution, a smart contract is translated into EVM code. Ethereum block execution time is approximately 15 seconds.
  The EVM, transactions, ether, consensus algorithm, gas, smart contract, and accounts are the core components of Ethereum. Ethereum stores the BC network's current state and a list of transactions. Moreover, the current state has details of Ethereum accounts, i.e., contract accounts, and externally owned accounts. These accounts contain information like a nonce, account EVM code hash, balance (ether), and storage root fields [36]. Ethereum uses ethash as a consensus mechanism. Furthermore, the Dthash consensus mechanism uses Keccak-256 and Keccak-512 hashing algorithms. Ethereum uses ethers to buy gas and run smart contracts on the EVM.

- Hyperledger: Hyperledger is an open-source project. It is developed by more than 100 companies in the industry. Hyperledger has different projects, i.e., Hyperledger Iroha, Hyperledger Fabric, Hyperledger Sawtooth etc.
  Hyperledger Fabric was developed by IBM under the Hyperledger project. It is a permissioned BC. It uses Javascript, GO, or Java language to develop distributed applications [37]. In Hyperledger, chaincode is a programmable code that runs on top of BC. The chaincode is utilized to produce resource definitions and business contracts. The Fabric network consists of three nodes i.e., ordering service nodes (OSN), peer nodes, and client nodes [38]. The OSN executes the consensus protocol. Similarly, the peer node executes chaincode whereas client nodes accept transaction proposals from the users. The Fabric executes a transaction in three steps [39].

- Endorsement Step: The endorsing peers accept transaction proposals from the clients. Moreover, it runs the chaincode and retrieves the current state of the ledger.

- Ordering Step: The ordering node accepts transactions from clients and produces a block of transactions.

- Validation Step: All peer nodes use endorsement policy i.e., Validation system chaincode to validate the transactions.

## 2.2 Related works

We have divided related works on authentication and authorization into three categories, i.e., access control Mechanisms for Internal users/vehicles (BC-based), Access Control Mechanisms for External users/vehicles (BC based), and Access Control Mechanisms for Internal/External users/vehicles (Centralized Trusted Mediator-based).

**2.2.1 Access control mechanisms for internal users/vehicles (BC-based).** In [40], a BC-based access control architecture is proposed by the authors. The owners generate access policies for their services and publish them in BC. These access policies are used to validate user requests for a resource. The proposed architecture allows users to use the service after authentication and access policy verification. The proposed architecture claimed that access policy verification and the rights assigning procedures are secure, transparent, and auditable. The study [41] proposed a secure framework for patient data transmission within healthcare systems. Patients initially register with the hospital server using personal and medical details, obtaining a special ID and certificate from the network manager for future communications. The Confidential Transmission Key Generation Module (CTKGM) generates a private-public key pair for encrypted data post-registration. Moreover, BC technology is integrated into the proposed architecture. It prevents unauthorized access to patient's private and confidential information. Similarly, the authors in [42] have proposed a BC-based authentication and authorization architecture. They have defined a single smart contract for BC. It uses access policies and takes authorization decisions. Furthermore, devices i.e., vehicles, and sensors are not included in the BC network. Therefore, these devices use a communication component called a "management hub" to run the smart contract.

These architectures authenticate and authorize users/patients/devices within a single domain. However, these architectures cannot authenticate or authorize cross-domain users/patients/devices. These frameworks are compared in Table 6.

**2.2.2 Access control mechanisms for external users/vehicles (BC based).** Authors, in [43], have proposed VeidBlock as a BC-based devices authentication mechanism for Software Defined Networks (SDN). In the proposed framework ViedBlock, i.e., Verifiable Identity

Block is used by devices to create BC-based identities. A local registration repository or local CA is used to authenticate SDN devices. After successful authentication, SDN devices get a VeidBlock from the Identity Provider Authority and Validator. The VeidBlock stores an anonymous identity for the SDN device. In [44], the authors have proposed a BC-based architecture for cross-platforms. The architecture is divided into two trust domains i.e., local, and global. The local devices form a local trust domain to share resources. Similarly, collaborating organizations form a global trust domain to share resources. The proposed architecture has used policies to build trustful relationships among member organizations. Similarly, in [45] a BlendCAC architecture is proposed for device authentication. The proposed architecture uses capability tokens to delegate or revoke permission on a particular object. A smart contract is implemented to utilize the capability tokens.

The authors, in [46, 47], have proposed a cross-domain access control architecture. In the proposed architecture, smart homes are connected with cluster heads. These cluster heads form the BC overlay network. The devices, installed in a smart home, have limited storage and computation power. Therefore, the proposed BC-based architecture has terminated coin and PoW without losing the security and privacy of the BC.

A legal device can exploit delegated rights. Therefore, user/device authentication is not sufficient. So, a mechanism is required to perform both platform verification and authentication before the user/device's authorization. Moreover, the design of these frameworks overlooked the importance of adhering to the separation of duty and least privilege principle. As a result, users/devices within the system might be granted excessive privileges. These frameworks are compared in Table 6.

**2.2.3 Access control mechanisms for internal/external users/vehicles (Centralized trusted mediator based).** In [9], the authors have proposed an authentication and authorization architecture. In the proposed architecture, cross-domain authentication and authorization are performed by a centralized mediator i.e., Delegation Service (DS). It sends the external user's request to his home domain for authentication. Similarly, it permits users to access the services after policy verification.

A capability-based access control architecture has been proposed in [48]. A centralized mediator transfers rights in the form of capability. A capability consists of a resource and a set of rights. During registration, SR gains a minimum set of permissions. Later on, more permissions are assigned to the SR according to the needs. Similarly, IoT-OAS [49] is an access control architecture for the Internet of Things. The SP sends access requests to the OAuth-based authorization service (OAS). The OAS is a centralized trusted service, based on stored policies that permit or deny access requests.

These architectures have targeted cross-domain authentication and authorization. However, user/device authentication and authorization are performed by a centralized trusted third entity. Therefore, the user/device's authentication and authorization will not happen, if the proposed centralized entity stops working. Moreover, the centralized trusted entity is exposed to different attacks, i.e., DoS, and DDoS. These frameworks are compared in Table 6.

## 3. Methods

The proposed BlockAuth architecture is a BC-based authentication and authorization architecture for ITS. The smart contract allows vehicles to obtain authentication from their home country. Then, the smart contract authorizes the vehicle based on access control policy validation. During authorization, the smart contract verifies the existence of an access control policy for the vehicle on the BC. If such a policy is identified, it then validates the prioritization of the policy with the least privilege. Additionally, it ensures that the vehicle is not granted two

contradictory policies from the same conflict of interest class. In the following sections, we discuss the BlockAuth architecture design and its formal modeling. Similarly, the BlockAuth architecture consists of five distinct system operations i.e., vehicle and RSU registration, region registration in BC, access control policy creation, access control policy revocation, and vehicle authentication and authorization.

## 3.1 Components of BlockAuth architecture

The BlockAuth architecture, as shown in Fig 1, consists of four core components i.e., Blockchain, Smart Contract, Blockchain Service Module, Road Side Unit (RSU), Regional Certification Authority (CA), and vehicle (On Board Unit).

**3.1.1 Blockchain network.** A BC network is structured as a mesh network comprised of CA nodes. After registration, each country deploys a CA node, integrating it into the BC

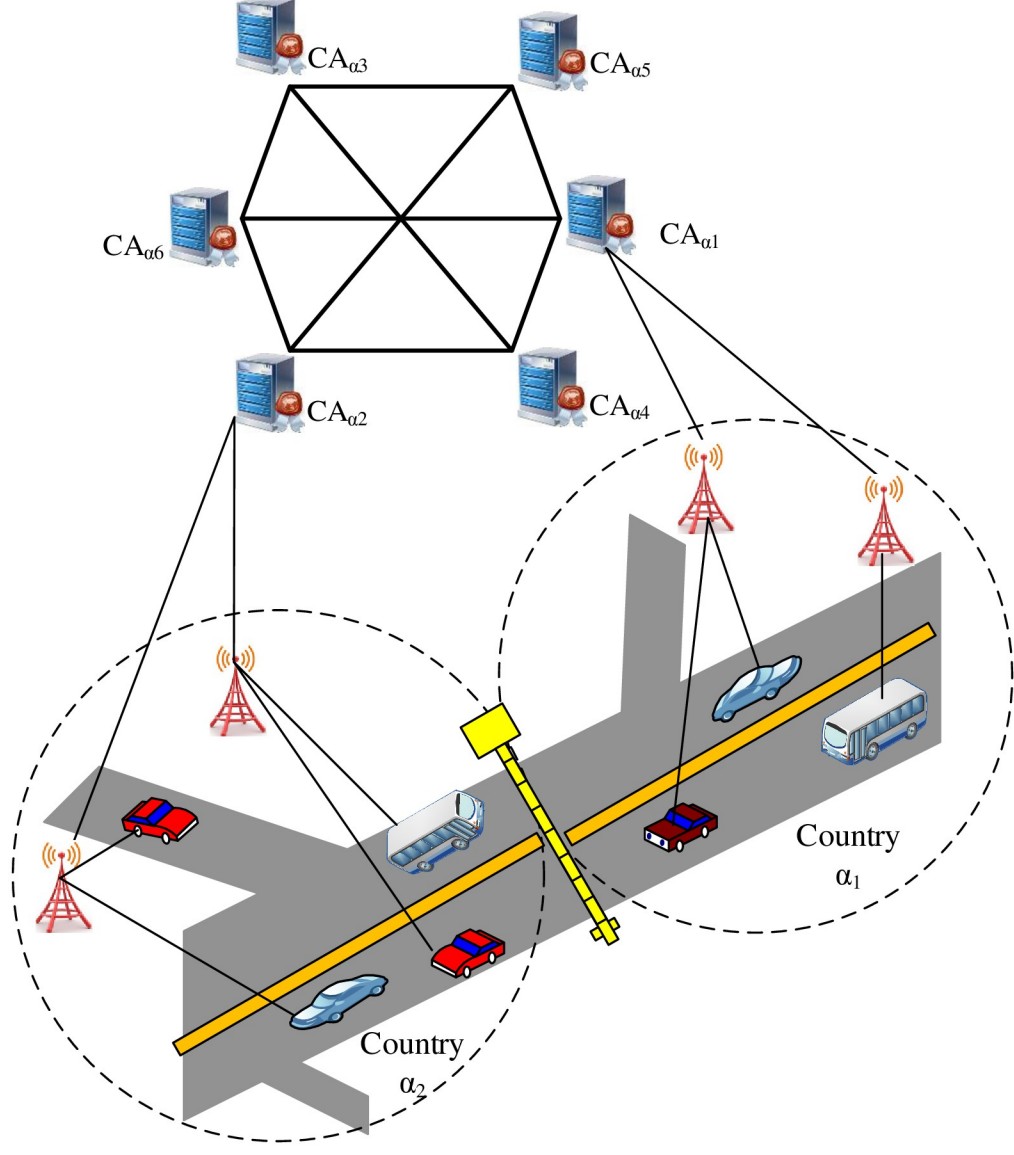

**Fig 1. High-Level architecture.**

**Table 1. Blockchain transactions and their descriptions.**

| | |
|---|---|
| "T.register" | To register user/vehicle, RSU and Country $\alpha$ in the BC. |
| "T.publish" | To load an access control policy in the BC. |
| "T.revoke" | To revoke an access control policy from the BC. |
| "T.join" | user/vehicle generates "T.join" transaction to join RSU region. |

network. Consequently, the BC network is segmented into distinct components i.e., CA nodes, BC storage, and smart contracts. The BC storage specifically holds cross-border access control policies designed for external vehicles. The BC network effectively creates a virtual coalition, uniting CA nodes from diverse countries.

**3.1.2 Smart contract.** The proposed smart contract implements cross-border vehicle authentication and authorization. The proposed BlockAuth architecture consists of a single smart contract.

**3.1.3 Blockchain service module.** The Blockchain Service Module (BSM) generates BC transactions, when it receives a request from a vehicle. Then, it forwards the BC transaction to the smart contract. The Table 1 contains BC transactions and their descriptions.

The details tasks of BSM are shown in Fig 2. The following are the functions of the proposed BSM.

**3.1.4 Road side unit.** RSUs are installed along highways that allow vehicles to communicate with BC.

**3.1.5 On-Board unit.** Every vehicle has an embedded processing unit called the On-Board Unit (OBU). The communication between vehicle and infrastructure (V2I) or vehicle and vehicle (V2V) is governed by OBU. The V2I and V2V wireless communication uses the DSRC (Dedicated Short Range Communications) protocol.

**3.1.6 Regional Certification Authority (RCA).** Every country has a centralized RCA. The RCA registers internal vehicles and RSUs. It generates Long Term certificates (LTC) for both vehicle and RSU after platform verification.

## 3.2 Operational processes within the BlockAuth

We assume a TPM module is installed in every vehicle and RSU. The following section contains a discussion of the core functions of the proposed framework.

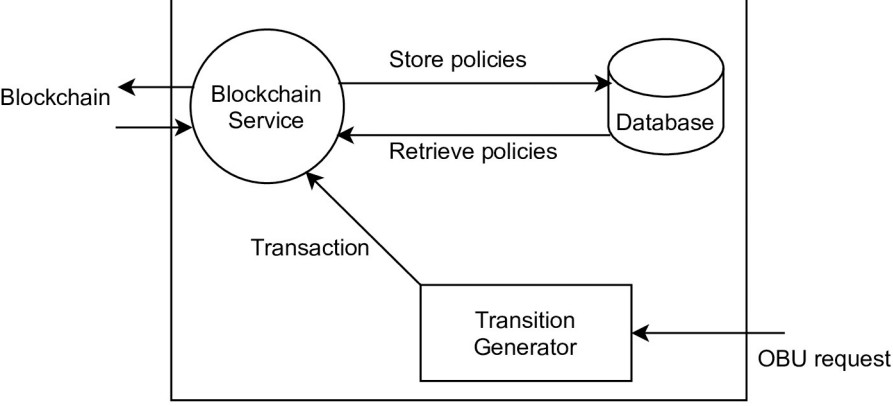

**Fig 2. Blockchain service module.**

- vehicle and RSU registration.

- Region registration in BC

- "Access control policy" creation.

- "Access control policy" revocation.

- vehicle Authentication and Authorization.

**3.2.1 Vehicle and RSU registration.** Initially, RSU and vehicle sent an offline request to the regional CA through a secure channel. The CA generates LTC for RSUs and vehicles after the platform hashes attestation from the manufacturer. After receiving LTC for RCA, both RSU and the vehicle generate a set of pseudonym IDs and register their pseudonym IDs with RCA. The vehicle and RSU registrations are shown in Fig 3. The following are the steps for vehicle and RSU registration.

- In step 1, the vehicle/RSU sends a registration request to the RCA.

- In step 2, RCA forwards the vehicle/RSU platform hash value to the manufacturer for attestation.

- In step 3, the manufacturer attests platform hash value and returns either a successful or unsuccessful response to the RCA

- In step 4, the RCA generates LTC and sends it to the vehicle or RSU.

- In step 5, the vehicle/RSU generates a set of pseudonym IDs for itself and sends it for registration to RCA for registration.

- In step 6, the RCA blinds vehicle/RSU pseudonym IDs with their LTC.

We assume that both the vehicle and RSU have TPM installed. They use TPM to generate public/private keys. Then, it applies a hash function to the public key and generates a set of pseudonym IDs.

**3.2.2 Country registration in BC.** Only the country administration can send an enrollment request to the BSM. The enrollment request consists of the country name, the IP address of the CA node, and other meta-data. Then, the BSM initiates a "T.register" transaction and transmits it to the smart contract. When the registration process has succeeded, the BC returns the address of the smart contract. The country registration in BC is shown in Fig 4. The following are the steps for country registration in BC.

- In step 1, the country administration sends an enrollment request to the BSM.

- In step 2, the BSM initiates a "T.register" transaction. Then, sends the transaction to the smart contract.

- In step 3, the smart contract enrolls the country in the BC and returns the smart contract ID to the BSM.

- In step 4, the BSM forwards the smart contract ID to the country administration.

**3.2.3 Access control policy publication.** The country administration defines access control policies for external vehicles. Access control policies are generated based on bilateral relations between the countries. The publications of access control policies on BC for external

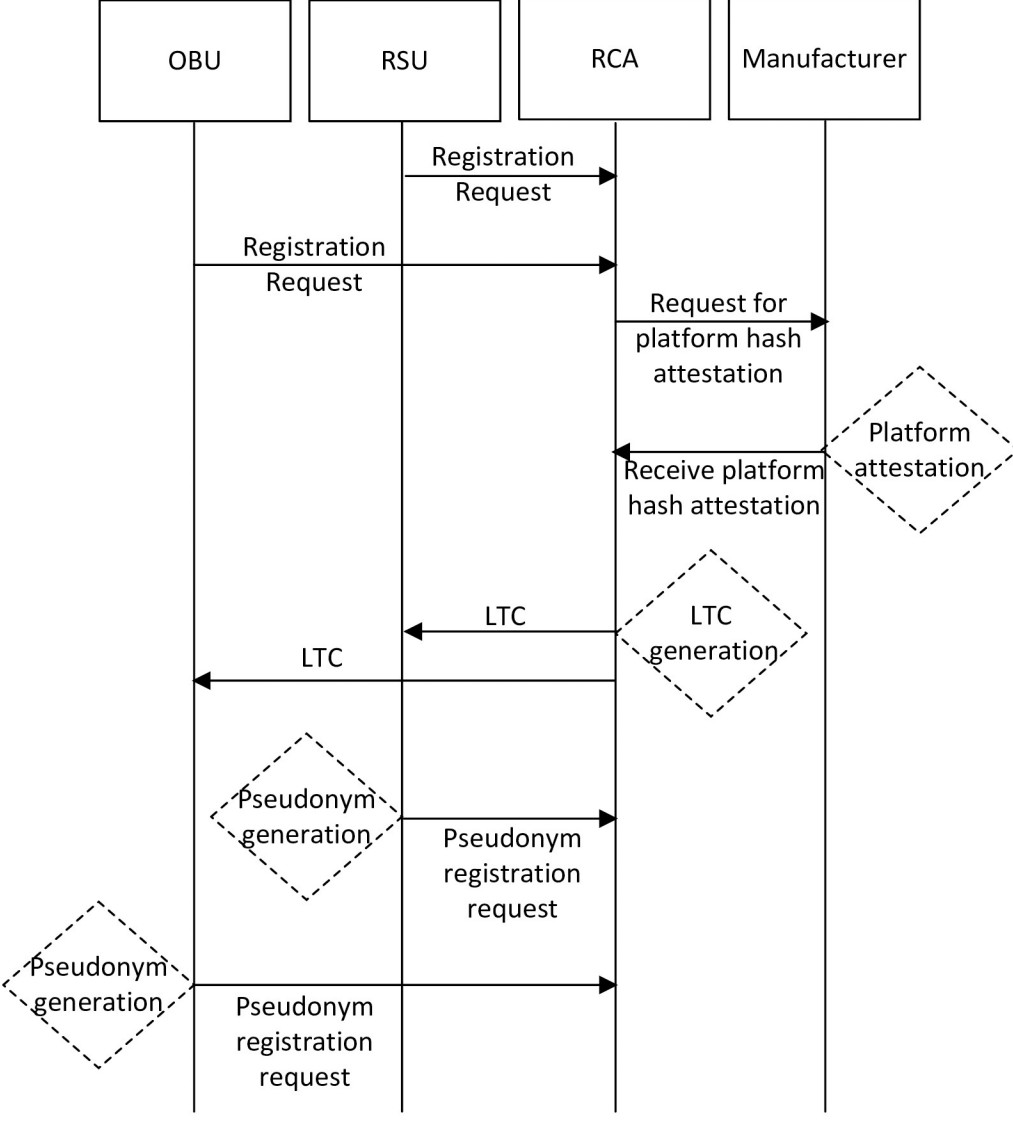

**Fig 3. Vehicle and RSU registration.**

vehicles are shown in Fig 5. The following are the steps in the publication of access control policies in BC.

- In step 1, the administrator sends an "access control policy" creation request to the BSM. Then, the BSM initiates a "T.publish" transaction.

- In step 2, BSM sends "T.publish" to the smart contract. Then, the smart contract validates the administrator's identity.

- In step 3, after successful authentication, the access control policy is saved in BC.

- In step 4, the smart contract returns a message i.e., "block added to the BC successfully" to the BSM.

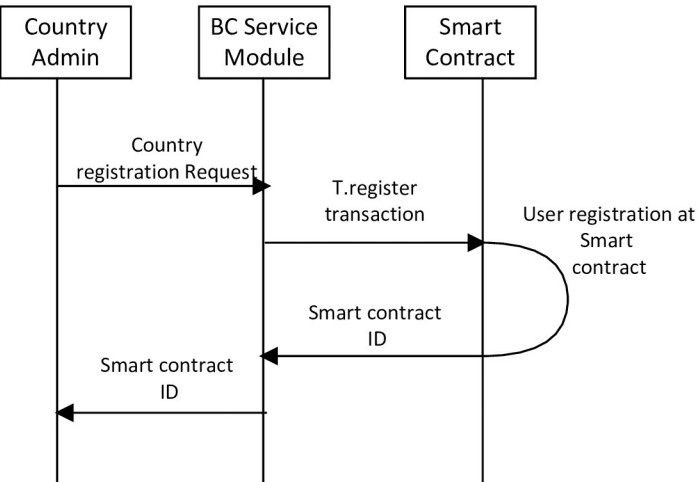

**Fig 4. Country registration.**

**3.2.4 Access control policy revocation.** The country's administration revoked access control policies for external vehicles. The revocation of access control policies from BC is shown in Fig 6. The access control policies revocation process consists of the following steps.

- In step 1, the administrator generates an "access control policy revocation request" and sends it to the BSM. Then, the BSM initiates a "T.revoke" transaction.

- In step 2, BSM sends "T.revoke" to the smart contract. Then, the smart contract authenticates the administrator.

- In step 3, after successful authentication, the access control policy is removed from the BC.

- In step 4, the smart contract returns a message i.e., "block removed successfully from the BC" to the BSM.

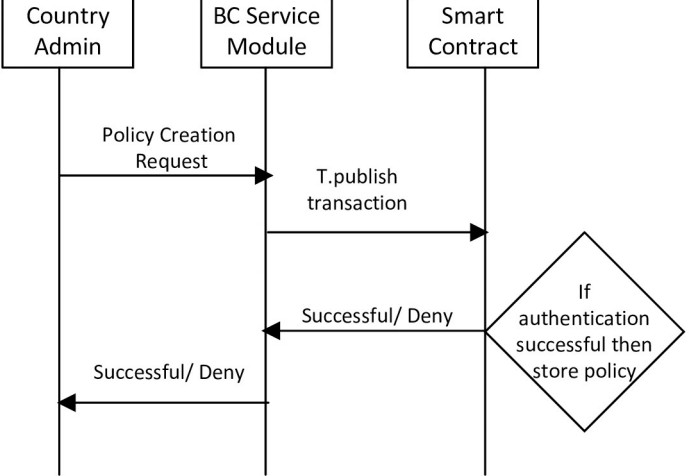

**Fig 5. Access control policy publication.**

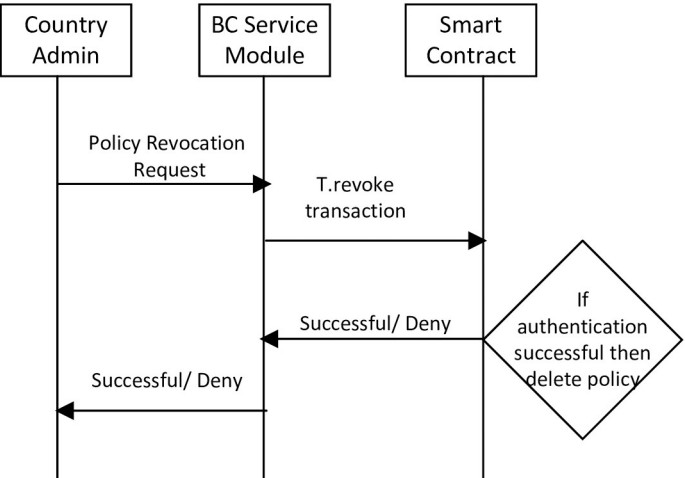

**Fig 6. Access control policy revocation.**

**3.2.5 Vehicle authentication and authorization.** The authentication and authorization of both external and internal vehicles are shown in Fig 7. The authentication and authorization process consists of the following steps.

- In step 1, the internal vehicle sends a join request to the RSU. It consists of vehicle pseudonym ID, vehicle platform hash value, and nonce and RCA signature. The join request is encrypted with a shared key used between RSU and the vehicle.

- In step 2, RSU decrypts the join request and forwards the vehicle pseudonym ID and RCA signature for validation to the RCA.

- In step 3, RCA verifies the signature associated with the vehicle pseudonym ID.

- In step 4, RSU allows the vehicle to join the network.

- In step 5, the external vehicle sends a join request to the RSU. Then, it forwards the vehicle's request to the BSM.

- In step 6, the BSM initiates a "T.join" transaction. Then, the BSM unicasts the "T.join" transaction to the smart contract. It uses platform hashes to authenticate internal users/vehicles based on platform hashes verification. Similarly, it authorizes vehicles based on access control policy validation.

- During steps 7a and 7b, the smart contract engages in vehicle pseudonym ID authentication by broadcasting to all RCAs across the BC network. These RCAs, located in various countries, each attempt to identify the Proof of Authenticity and Integrity (PoAI) for the vehicle. PoAI serves as a process that takes the vehicle pseudonym ID and yields the corresponding vehicle platform hash. However, the vehicle's home country RCA possesses knowledge of this hash. Subsequently, the country RCA transmits the PoAI to the smart contract.

- In step 8, the smart contract compares the platform hash value retrieved from the vehicle's home country RCA with the hash value obtained from the vehicle's request.

- In step 9, when the hash values match, the smart contract retrieves access control policies from the BC. Subsequently, it validates the vehicle's request against these policies. Initially, it confirms the presence of an access control policy for the requesting vehicle. If found, it then

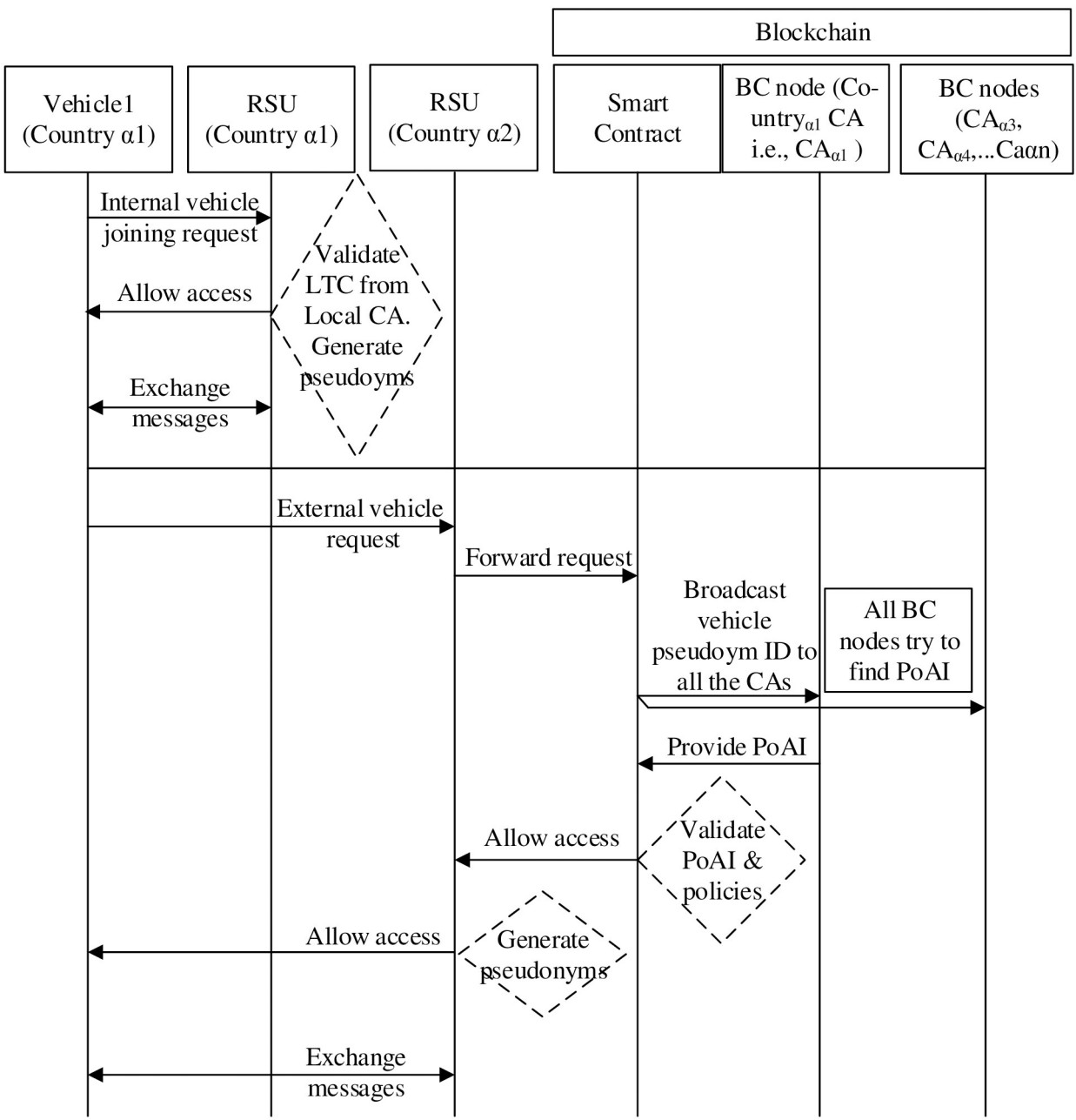

**Fig 7. Sequence diagram of the proposed framework.**

verifies the prioritization of the policy with the least privilege. Likewise, it ensures that the vehicle is not granted two conflicting policies from the same class of interests.

- In step 10, the smart contract generates an allow/deny message to the BSM.

- In step 11, the BSM forwards the authorization decision to the RSU.

- In step 12, based on smart contract authorization, the RSU allows/denies vehicle join requests.

## 3.3 Formal modeling of BlockAuth architecture

In this section, we formally present the core parts and system operations of the BlockAuth architecture.

**3.3.1 Notations.** The notations used in this section are given in Table 2.

**3.3.2 Components of the proposed model.** In the following subsections, the components of BlockAuth architecture are defined formally.

1. **Subject Set:**

$$\mathcal{S} = \Sigma_{i=0}^{\infty} s_i \tag{1}$$

$$s_i = user \mid vehicle \mid RSU \tag{2}$$

$\mathcal{S}$ is a superset of all the users, vehicles, and RSUs that generate requests for joining the network. $s_i$ can be a delegator if it delegates permission on his resource to $s_j$. Similarly, it can act as a delegatee if it receives permission on an object from $s_j$. Both the delegator and delegate are members of the subject set. A delegator provides permission for a resource, whereas the delegate receives the permission.

2. **Object Set:**

$$\mathcal{O} = \Sigma_{j=0}^{n} o_j \tag{3}$$

$\mathcal{O}$ is a superset of objects. It contains the entire data objects and services in BlockAuth architecture.

3. **Permission Set:**

$$\mathcal{P} = \Sigma_{j=0}^{n} p_k \tag{4}$$

$\mathcal{P}$ is a superset of all the permissions in BlockAuth architecture. We define permission as an operation on the object, i.e., $(o_i, op)$, where $o_i$ is the ith object and "op" is an operation.

**Table 2. Notations and their descriptions.**

| Notation | Description |
|---|---|
| $s_j$ | jth subject. |
| $o_i$ | ith object. |
| $p_m$ | mth permission. |
| $\cup_{i=0}^{\infty} s_i$ | all the subjects in the system. |
| $\cup_{j=0}^{\infty} o_j$ | all the objects in the system. |
| $\cup_{k=0}^{\infty} p_k$ | all the permissions in the system. |
| $\mathcal{LVS}$ | Set of all internal users/vehicles in a country. |
| $\mathcal{GVS}$ | Set of all external users/vehicles from other countries. |
| $\mathcal{ACPIV}$ | Set of all access control policies for internal users/vehicles. |
| $\mathcal{ACPEV}$ | Set of all access control policies for external users/vehicles. |
| $\{s_m{}^{p_q}s_n\}$ | Subject $s_m$ transferred a permission $p_q$ to object $s_n$. |
| $\{s_m \leftrightsquigarrow^{p_q} s_n\}$ | Subject $s_m$ revoked a permission $p_q$ from Subject $s_m$. |
| $\rho\hbar_{ca}$ | fresh calculated hash of user/vehicle platform. |
| $\rho\hbar_{st}$ | stored hash of user/vehicle platform in BC. |

4. **Local/Internal Vehicle Set:**

$$\mathcal{LVS}_k = \Sigma_{i=0}^n s_i \tag{5}$$

$\mathcal{LVS}_k$ is a superset of all the users/vehicles registered in a country $k$. Moreover, $\mathcal{LVS}_k \subseteq \mathcal{S}$, Local Vehicle Set is a subset of Subject Set.

5. **Global/External Vehicle Set:**

$$\mathcal{GVS}_k = \Sigma_{j=0}^n s_j \tag{6}$$

$\mathcal{GVS}_k$ is a superset of all the users/vehicles from other countries who entered into country $k$. Moreover, $\mathcal{GVS}_k \subseteq \mathcal{S}$, Global Vehicle Set is a subset of Subject Set.

6. **Access Control Policy:** Delegation policy ($\mathcal{ACP}$) is defined as a triple ($s_m$, $s_n$, $p_q$), where $s_m$ = delegator, $s_n$ = delegatee, and $p_q$ = set of permissions.

$$(s_l, s_m, p_q) \in \{s_l \leadsto^{p_q} s_m\} \tag{7}$$

7. **Country Domain:**

$$\mathcal{CD}_i = \Sigma_{m=0,n=0,q=0}^r \{s_m \cup o_n \cup p_q \cup \mathcal{DS}_\mathcal{L}\} \tag{8}$$

$\mathcal{CD}_i$ is a country domain set that contains all the subjects, objects, and rights.

8. **Pseudonymous IDs:**

$$\mathcal{PID} = \Sigma_{i=0}^n \mathcal{PID}_i \tag{9}$$

$\mathcal{PID}_i$ set contains the Pseudonymous IDs of vehicles and RSUs belonging to a single country.

9. **Access Control Policy Set for Internal Vehicles:**

$$\mathcal{ACPSIV} = \Sigma_{m=0,n=0,q=0}^r \{s_l \leadsto^{p_q} s_m\} \tag{10}$$

$\mathcal{ACPSIV}$ contains the total set of access control policies for internal users/vehicles within a country.

10. **Access Control Policy Set for External Vehicles:**

$$\mathcal{ACPSEV} = \Sigma_{m=0,n=0,q=0}^r \{s_m \leadsto^{p_q} s_n\} \tag{11}$$

$\mathcal{ACPSEV}$ contains the total set of access control policies for external users/vehicles.

11. **Platform Hashes Set:**

$$\mathcal{PHS} = \Sigma_{j=0,status=ca,st,re}^n \{\rho \hbar_{s_j,status}\} \tag{12}$$

$\mathcal{PHS}$ contains platform hashes of all the users/vehicles and RSUs within a belong to a single country.

12. **Virtual Coalition:**

$$\mathcal{VC} = \Sigma_{j=2}^r CD_j, \ni \{s_m \in CD_j\}, \{o_l \in CD_j\} \, and \{s_m \leadsto^{p_q} s_n\} \in \mathcal{ACPSEV} \tag{13}$$

$\mathcal{VC}$ contains all the member countries in the virtual coalition. A virtual coalition contains

two or more countries such that a vehicle in one country is allowed to move to another country without re-enrollment with CA.

13. **Axiom 1:** Every vehicle possesses a distinct platform hash value, ensuring that no two vehicles share the same value.

$$s_i \neq s_j \Rightarrow \rho\hbar_{st} \neq \rho\hbar_{st} \tag{14}$$

14. **Axiom 2:** If an attacker manages to infiltrate a vehicle's on-board units and installs malicious software, the resulting platform hash value will differ from the previously stored value. Consequently, the vehicle would be unable to successfully authenticate.

$$s_i = s_j \wedge \rho\hbar_{rec} \neq \rho\hbar_{st} \Rightarrow s_i \, is \, malicious \tag{15}$$

15. **Axiom 3:** In the case of multiple policies for a vehicle, the policy with the least privilege will take precedence.

$$\{s_i \in (\mathcal{ACP}_i \wedge \mathcal{ACP}_j)\} \wedge (\mathcal{ACP}_i \subseteq \mathcal{ACP}_j) \Rightarrow \mathcal{ACP}_i \tag{16}$$

16. **Axiom 4:** This axiom establishes the concept of separation of duty. When two policies present conflicting interests, the vehicle will grant authorization based on one of the policies.

$$\{(\mathcal{ACP}_i \wedge \mathcal{ACP}_j), 2\} \Rightarrow (s_i \in \mathcal{ACP}_i) \vee (s_i \in \mathcal{ACP}_j) \tag{17}$$

**3.3.3 Smart contract: Authentication and authorization procedure.** In the BlockAuth architecture, a single smart contract is in charge of executing cross-border authentication and authorization procedures. During authentication, this smart contract commences vehicle pseudonym ID validation by broadcasting it to all CAs. After obtaining the platform hash from the vehicle's parent CA, the smart contract compares the received platform hash in the vehicle's request with the one received from the CA. When a hash match occurs, the smart contract confirms the presence of an access control policy for the requesting vehicle on the BC. Subsequently, it undertakes the verification of adherence to the separation of duty and least privilege rules. The **Algorithm. 1** outlines the functioning of the proposed smart contract.

**Algorithm 1** Smart Contract Operations

```
1: Input: T.join (s_j, ρℏ_ca)
2: Output: allow or deny
3: if (s_j ∈ LVS_k) then
4:   validates LTC signature from local CA and platform hash from
local Registration Authority.
5:   if (ρℏ_ca ≡ ρℏ_st) then
6:     generate pseudonyms ID PID
7:   else if (ρℏ_ca ≠ ρℏ_st) then
8:     deny
9:   end if
10: else if (s_j ∈ GVS_k) then
```

```
11:    send the signed pseudonymous ID Sig{PID_{s_j}}_{SK_{countryβ1}} of the request-
ing vehicle to all the BC peers
12:    Receive signed platform hash value Sig{ρℏ_s}_{SK_{countryα1}} of the request-
ing vehicle from his country.
13:    while (NOT got first PoAI from the peers) do
14:      if (ρℏ_{ca} ≡ ρℏ_{st}) then
15:        fetch access control policy from BC.
16:        if {{(s_m⤳^{p_q} s_j) ∈ ACPSEV} then
17:          Policy exist
18:          if {(s_m⤳^{p_q} s_j) ⊇ (s_m⤳^{p_r} s_k)} then
19:            (s_m⤳^{p_r} s_k) Policy with least privilege selected
20:          else
21:            (s_m⤳^{p_q} s_j) Only one policy exist.
22:          end if
23:          if {(s_m⤳^{p_q} s_j) ∧ (s_m⤳^{p_r} s_k)}, 2} = 0} then
24:            Allow request (No conflict of interest among policies).
25:          else
26:            Deny request (Conflict of interest exists).
27:          end if
28:        else
29:          Deny request (No policy exists).
30:        end if
31:      else if (ρℏ_{ca} ≠ ρℏ_{st}) then
32:        Deny request (malicious request)
33:      end if
34:    end while
35: end if
```

**3.3.4 Proof-of-Authenticity and integrity.**   The proposed smart contract uses PoAI to validate the authenticity and integrity of the external vehicle. The BC broadcasts external vehicle pseudonymous IDs to all the CA peers in the virtual coalition. The CAs try to resolve the vehicle's pseudonymous ID to his platform hash called PoAI. **Algorithm. 2** describes the calculation of PoAI.

**Algorithm 2** Proof-of-Authenticity and Integrity

```
1: Input: PID_{s_j}
2: Output: Sig{ρℏ_{st}}_{SK_{countryα1}}
3: Receive broadcast from the BC network with signed pseudonymous ID
Sig{PID_{s_j}}_{SK_{countryβ1}}.
4: Verify the signatures and resolve Pseudonymous ID to platform hash.
5: if (PID_{s_j} resolved to ρℏ_{st}) then
6:    return Sig{ρℏ_{st}}_{SK_{countryα1}}
7: else if (PID_{s_j} Not resolved to ρℏ_{st}) then
8:    return error
9: end if
```

**3.3.5 Access control policy creation.**   The "Access Control Policy Creation" algorithm is presented in Algorithm 3. Initially, algorithm 3 accepts policy creation requests from the administrator. In response, it generates an error if the policy exists in the BC. Otherwise, it creates an access control policy and stores it in BC. In lines 3-7, if the requester is an internal vehicle, and there is no matching access control policy exit. Then, Algorithm 3 adds an access control policy in BC. Similarly, in lines 9-13, Algorithm 3 adds an access control policy for the external vehicle when the requester is an external vehicle, and there is no matching access control policy exit.

**Algorithm 3** Access Control Policy Creation Algorithm

```
1: Input: T.publish(s_m, s_n, p_q)
```

```
2: Output: {s_m ⤳^{p_q} s_n} or error
3: if (s_n ∈ LVS_k) then
4:    if {s_m ⤳^{p_q} s_n} ∈ ACPSIV then
5:       return error (Duplicate policy)
6:    else
7:       ACPSIV = ACPSIV + {s_m ⤳^{p_q} s_n}
8:    end if
9: else if (s_n ∈ GVS_k) then
10:    if {s_m ⤳^{p_q} s_n} ∈ ACPSEV then
11:       return error (Duplicate policy)
12:    else
13:       ACPSEV = ACPSEV + {s_m ⤳^{p_q} s_n}
14:    end if
15: else
16:    return error
17: end if
```

**3.3.6 Access control policy revocation.** The "Access Control Policy Revocation" algorithm is presented in Algorithm 4. Algorithm 4 accepts policy revocation request from the administrator and removes the policy from the BC. Similarly, it generates an error if the policy is not available in the BC. In lines 3-5, if the delegatee is an internal vehicle, and the access control policy is not available in the BC. Then, it generates an error. In lines 6-8, it revokes the access control policy from the BC. Similarly, in lines 9-11, if the delegatee is an external vehicle, and the access control policy is not available in the BC. Then, it returns an error. In lines 12-14, Algorithm 4 revokes the access control policy from the BC.

**Algorithm 4** Access Control Policy Revocation Algorithm

```
1: Input: T.revoke(s_m, s_n, p_q)
2: Output: {s_m ⤳^{p_q} s_n} or error
3: if (s_n ∈ LVS_k) then
4:    if {s_m ⤳^{p_q} s_n} ∉ ACPSIV then
5:       return error (Policy does not present)
6:    else
7:       ACPSIV = ACPSIV − {s_m ⤳^{p_q} s_n}
8:    end if
9: else if (s_n ∈ GVS_k) then
10:    if {s_m ⤳^{p_q} s_n} ∉ ACPSEV then
11:       return error (Policy does not present)
12:    else
13:       ACPSEV = ACPSEV − {s_m ⤳^{p_q} s_n}
14:    end if
15: else
16:    return error
17: end if
```

## 3.4 Compatible usecase

Suppose vehicle "A" moves from country $\alpha_1$ to country $\alpha_2$. A vehicle "A" sends a "join" request to the nearby RSU as shown in Fig 8. The "join" request contains vehicle pseudonym ID, platform hash value, nonce, and RCA signature. Then, the RSU forwards the vehicle pseudonym ID and RCA signature for validation to the smart contract. The smart contract authenticates the vehicle pseudonym ID by sending a broadcast to all the verifier nodes in the BC network. In response, the CA node of the vehicle's home country sends the PoAI. Subsequently, the smart contract performs a comparison of the hash values received in PoAI and vehicle requests. If both hash values are equal, the smart contract proceeds to retrieve access control policies from BC. Then, it validates the vehicle's request against the access control policies.

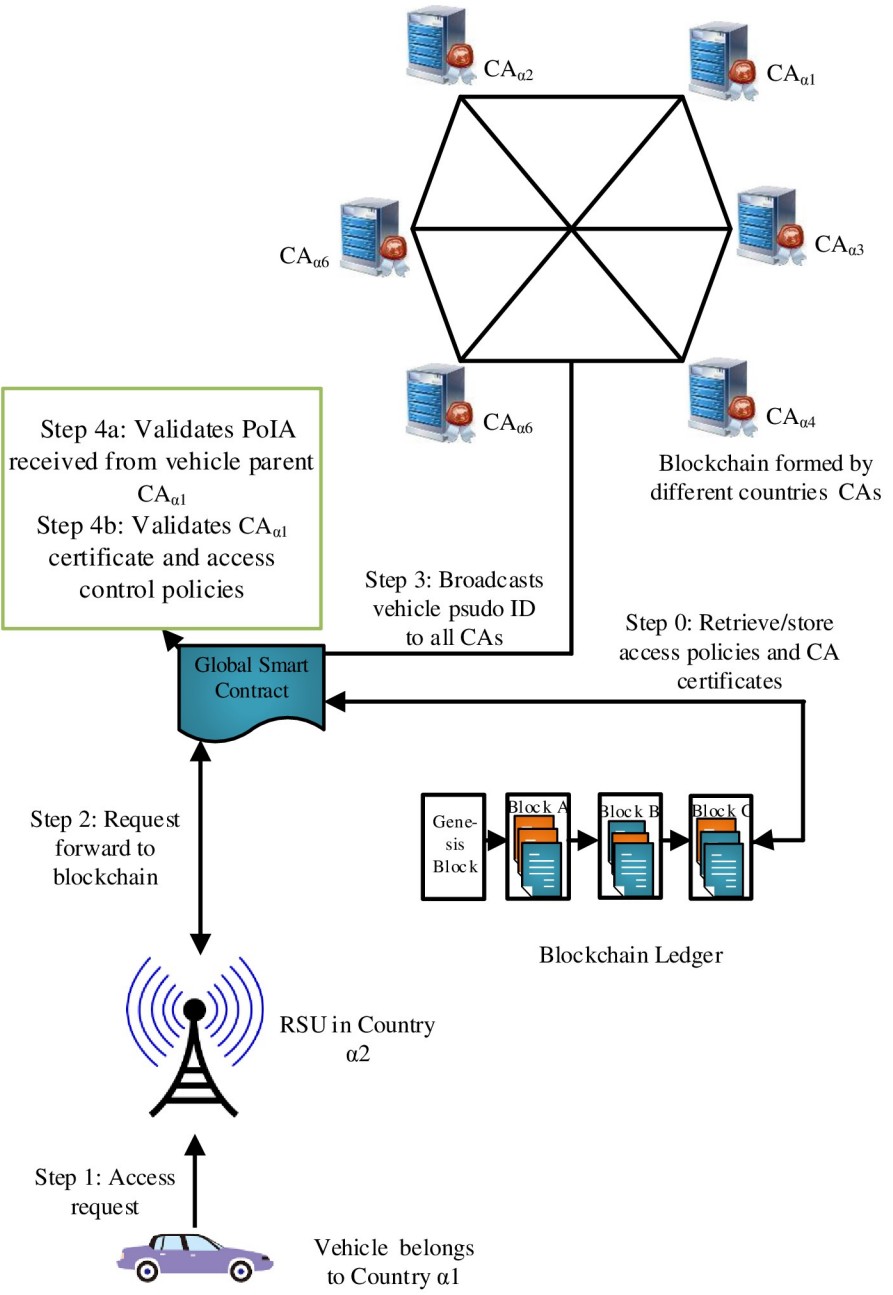

**Fig 8. Usecase.**

Finally, the RSU allows/denies vehicle join requests based on the authorization decision of the smart contract.

## 4 Results and discussion

In this section, we discuss the implementation of the proposed framework in Node.js. Additionally, we discuss performance evaluation, security analysis, and comparison with existing similar architectures.

## 4.1 Implementation details

We have implemented the BlockAuth architecture in Node.js. Our implementation consists of a client application, server nodes, database node, chaincode, and BC network.

**4.1.1 Client application.** A third-party application, i.e., Postman, is used for communication with the BC.

**4.1.2 Server nodes.** Our experimental setup consists of a pair of server nodes representing two distinct SP countries. These server nodes receive client requests through specific ports and subsequently call the chaincode for authentication and authorization.

**4.1.3 Database node.** A Database node is used to implement MongoDB. MongoDB stores access control policies, user/vehicle Pseudonymous ID, and platform hashes.

**4.1.4 Chaincode.** A chaincode, also called a smart contract, enforces business logic. Our implemented chaincode performs cross-border vehicle authentication and authorization. This process involves redirecting an external vehicle's pseudonymous ID to all relevant RCAs. Subsequently, it obtains PoAI from the vehicle parent RCAs. Afterward, it compares the platform hash values associated with the vehicle request and the PoAI response. Once the vehicle authentication is successfully completed, the chaincode proceeds to allow or deny the vehicle based on the access control policies stored on the BC.

It broadcasts an external vehicle pseudonymous ID to all the RCA. In response, it receives PoAI from RCA. Then, it validates the vehicle platform hash values received in the vehicle request and PoAI response. The vehicle is verified if both the hash values are similar. If the vehicle is authenticated successfully, then the chaincode validates access control policies stored in BC. Therefore, the vehicle is either allowed/denied based on access control policies.

**4.1.5 BC network.** In the BC network, the initial block of the chain is known as the genesis block. In the chain, every block has a hash value of the previous block except the genesis block. The proposed block design is given in Table 3. Similarly, the implemented BC network configurations are given in Table 4.

## 4.2 Performance evaluation

In this section, we carry out chain size analysis, throughput analysis, and overhead ratio analysis.

**4.2.1 Chain size analysis.** To measure the execution time of each stage, i.e., authentication, authorization, policy publication, and policy revocation, 500 test executions have been conducted. We followed the scenario illustrated in Fig 9 for conducting these experiments. In the proposed scenario, an external vehicle initiates an access request and sends it to the smart contract for authentication and authorization. We evaluate the BlockAuth architecture with 1, 100, 1000, and 5000 access control policies shown in Fig 9. Moreover, the number of virtual

**Table 3. Block design.**

| | |
|---|---|
| perviousHash: | "dfbd245030921ef6dff8eb1a9a90f6ba978037c268166d89d1f0603fd38f9e861" |
| delegator: | "Alice" |
| delegatee: | "Bob" |
| vehicleId: | "vehicle218" |
| Permission: | "read" |
| validTill: | "28-April-2023" |
| hash: | "6fcd393bea9d457d2bcb26c5cfa8931894ab6f097c8c98b47a55fe8f03cc4cb7" |
| timestamp: | "Wed, 12 April 2023 10:45:57 GMT" |

**Table 4. Network configuration.**

| Parameters | Values |
|---|---|
| Node Resources | CPU Speed = Intel(R) Core(TM) i7-8750H CPU @ 2.20 GHz RAM = 32 GB |
| Database | MongoDB |
| Block Size | 571 byte |
| Network Speed | 100 Mbps |

requests was maintained at a constant 400. The graph shows that the increment in policy count has no significant impact on the execution time of the authentication procedure. Similarly, the time taken for policy creation does not notably influence an increase in policy publication duration. However, the execution time for authorization and policy revocation procedures shows an observable increase as the number of policies rises. This is attributed to the extended time required for searching through an enlarged chain of policies.

**4.2.2 Throughput analysis.** We analyze the operations of BlockAuth architectures with concurrent vehicle requests. However, we tested the BlockAuth architecture with a different number of virtual vehicle requests, i.e., N = 50, 100, 1000, 5000. Moreover, no. of access control policies = 500 (constant). At first, the experiment was tested for 50 concurrent virtual vehicle requests. The authentication operation completes in 240ms, authorization operation completes in 30ms. Moreover, access control policy creation and revocation are completed in 120ms and 87ms respectively. Similarly, the test is performed for n = 200, 400, 600, and 800 concurrent vehicle requests. The outcomes are given in Fig 10.

**4.2.3 Overhead ratio analysis.** An external vehicle sends a "T.join" transaction for registration with RSU in another country. The BlockAuth architecture allows the requesting vehicle to access the service after PoAI and policy verification. Similarly, the administrator uses the "T.publish" transaction to create a new access control policy. Similarly, the "T.revoke" transaction is used to delete the "access control" policy. Furthermore, the administrator uses the "T. register" transaction to enroll vehicle, RSU, and a country $\alpha$ in the BC network.

Every session contains a number of security operations, i.e., encryption, decryption, a hashing function, signature creation, and signature confirmation. Fig 11 shows the execution time

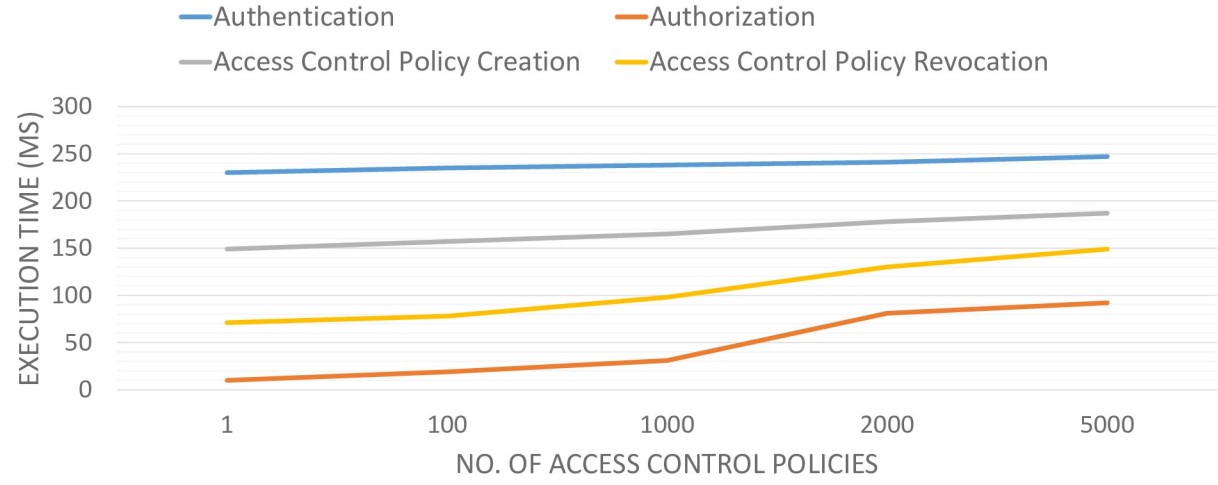

**Fig 9. Chain size analysis.**

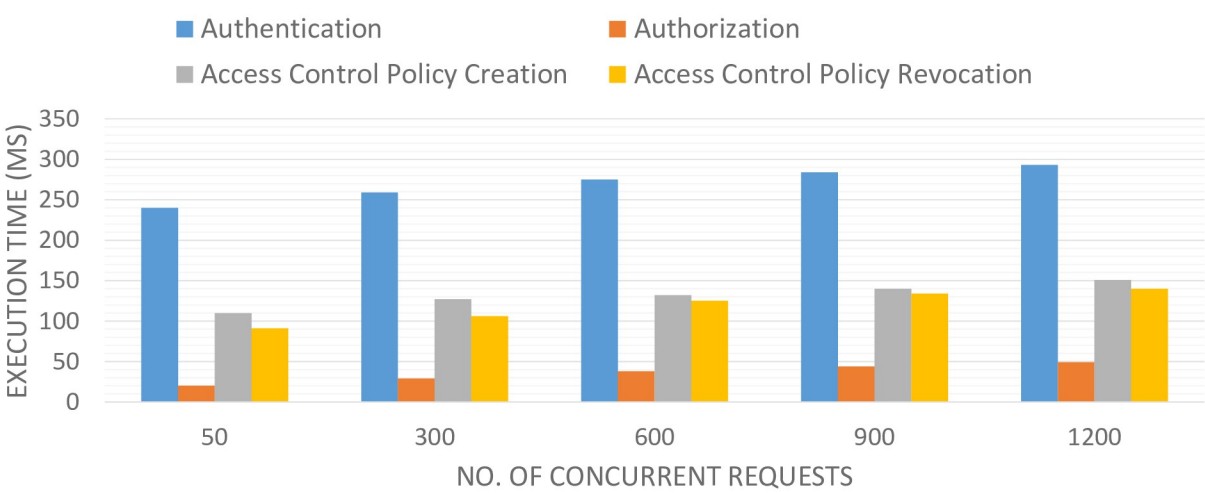

**Fig 10. Execution times of block hashing, transaction encryption, and PoIA.**

of all the transactions with/without encryption. We keep the number of vehicle requests and "access control" policies constant i.e., N = 200 and P = 1000. The overhead is negligible, i.e., less than 2%.

## 4.3 Security analysis and threat models

The following subsection consists of security analysis and threat modeling of the proposed framework.

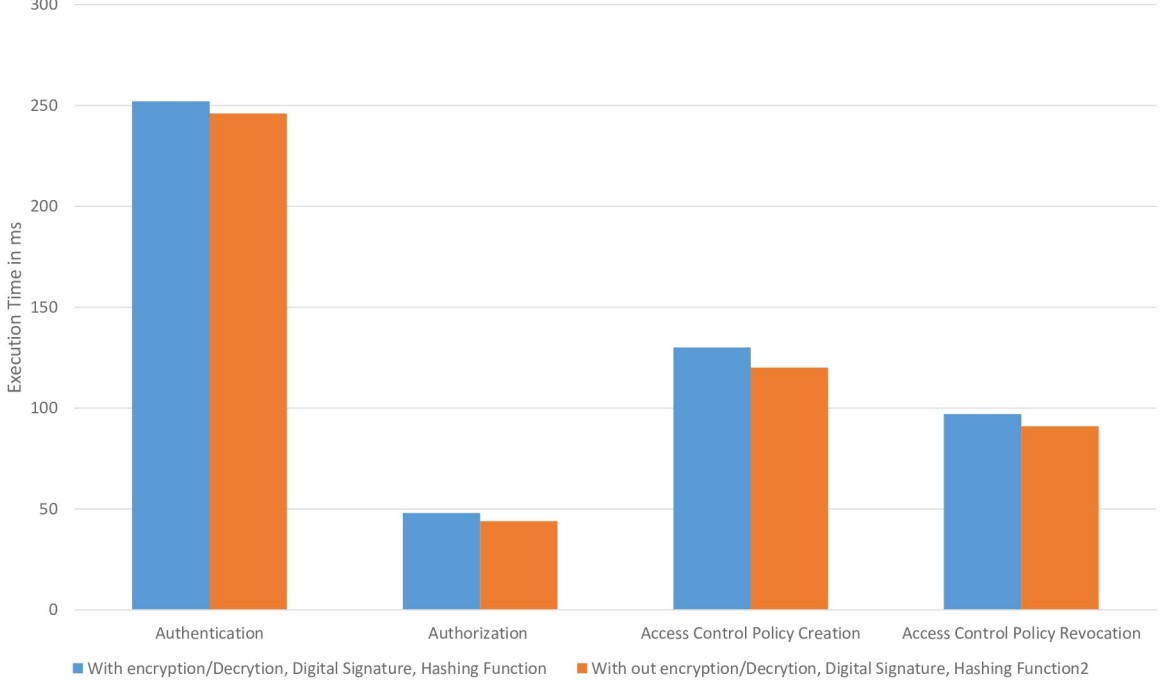

**Fig 11. Comparison of different transactions with/without encryption, digital signature, and hashing function.**

**4.3.1 Security analysis.** We use the CIAAN model to analyze the security of the Block-Auth architecture. We use BC to fulfill the security needs of the CIAAN model. The summarized analysis of the BlockAuth architecture is given in Table 5.

- **Confidentiality:** It means to ensure that the data is not accessed by illegal users/vehicles and RSUs. Public-key encryption is used in the BlockAuth architecture to make communication among vehicles, RSU, and BC reliable.

- **Integrity:** It means to protect data from improper modification. The proposed framework uses integrity measurements provided by BC. The BC, SHA-256 cryptographic hash function is used to protect the vehicle and RSU platform data as well as data during communication.

- **Availability:** It means to ensure that resources are accessible to users/vehicles and RSUs when required. The BC ledgers have a built-in replication mechanism. Therefore, a damaged node can restore its ledger's data from other peer nodes. Furthermore, the proposed PoAI mechanism ensures system availability against DoS and DDoS attacks.

- **Authentication and Authorization:** It means to ensure that the vehicle and RSU are legitimate. The proposed PoAI mechanism allows users/vehicles to get authentication from their parent country. Similarly, the proposed framework authorizes vehicles based on access control policies stored in the BC.

- **Non-repudiation:** It means to ensure that the users/vehicles can not refuse their committed transactions. In the BlockAuth architecture, cryptographic keys are used to attest users'/vehicles' transactions. Hence, the digital attestation ensures non-repudiation for user/vehicle transactions.

**4.3.2 Threat models.** The following are the potential threats to the proposed BlockAuth framework.

- **Malicious User/Vehicle:** Suppose a scenario where a malicious user/vehicle wants to get authentication by spoofing a legitimate user/vehicle ID. In the proposed framework "T.join" request consists of the user/vehicle platform hash and pseudonymous ID. The smart contract verifies the user's platform hash from his country's RCA. The RCA binds user/vehicle pseudonymous IDs with the user's platform hash during initial registration. As a result, it thwarts any attempt to initiate a spoofing attack.

- **Malicious Service Provider:** Suppose a scenario where a malicious SP intends to uncover the true identity of a user/vehicle. In the proposed framework users/vehicles provide their

**Table 5. Evaluation of security parameters.**

| Parameters | Description |
|---|---|
| Confidentiality | BlockAuth architecture uses asymmetric encryption to make connections among OBUs, RSUs, and smart contracts safe. |
| Integrity | BC built-in function SHA-256 is used for data and platform integrity preservation. |
| Availability | BC ledgers replication mechanism is used to achieve data availability. |
| Authentication/ Authorization | Proof of Authenticity and Integrity is used to authenticate OBU. Similarly, access control policies help in cross-border OBUs authorization. The smart contract is designed to restrict user activities by ensuring separation of duty and least privilege principles. |
| Non-repudiation | To achieve non-repudiation, every OBU and RSU digitally signed his transaction. |

platform hashes and pseudonymous IDs to the SP for the purpose of authentication. The SP subsequently redirects the users/vehicles to their respective country's RCA for authentication, because it stores the real identity of the user/vehicle. As a result, the country RCA provides the stored platform hash of the user/vehicle. Therefore, it is impossible for SP to extract the genuine identity of the user/vehicle from the platform hash.

- **DoS/DDoS Attack on Service Provider Authentication Service:**
  Suppose a scenario where an attacker uses a compromised user/vehicle platform and initiates a DoS/DDoS attack to overwhelm the services of the SP. However, the compromised user/vehicle's current platform hash must be different from the hash stored with the user/vehicle's country RCA. This disparity allows SP to deny the malicious access request during the authentication process. As a result, this countermeasure prevents the attacker from initialing a DoS/DDoS attack.

## 4.4 Comparative analysis

The Comparisons among BlockAuth and existing related architecture are given in Table 6.

## 5. Conclusion

We proposed a BC-based cross-border vehicle authentication and authorization architecture. The proposed BlockAuth architecture consists of a BC network of CA nodes. The BC network authenticates and authorizes cross-border vehicles. The substitution of the centralized access control service outlined in the existing literature with BC technology mitigates the vulnerability that could result in system failure and various cyber-attacks like DoS and DDOS. Similarly, storing access control policies on the BC ensured the prevention of illegal authorization, thus enhancing the security and integrity of the framework. Unlike existing frameworks, the authorization service is enforced with separation of duty and least privilege principles to ensure that users are not granted more privileges than their necessary needs. Moreover, unlike conventional frameworks, BlockAuth allows user/vehicle authentication from his home country CA. Thus, ensures the protection of user/vehicle credentials. Furthermore, Hyperledger is used to implement the BlockAuth architecture. The BlockAuth ensures secure vehicle authentication and authorization with high throughput and minimal computational overhead, under 2%.

### 5.1 Limitations of the proposed BlockAuth framework

The following are the limitations of the BlockAuth framework.

- Scalability: As more CAs become part of the virtual coalition, the count of BC nodes within the overlay network will grow. This expansion increases computational overhead. As a result, the BlockAuth exhibits limited scalability.

- Storage: The proposed BlockAuth stores access control policies on BC. However, these policies consume a larger amount of storage due to their complex structure.

### 5.2 Future directions

We outline potential avenues for future research exploration.

- Security: Future research could involve BlockAuth resilience against emerging attacks on BC. Additionally, BC security could be enhanced by using formal modeling and verification techniques.

**Table 6. Comparisons among porposed and exiting frameworks.**

| Reference No. | Access control Mechanisms for Enternal users/vehicles (BC-based) | Access control Mechanisms for Internal users/vehicles (BCbased) | Access control Mechanisms for Internal/External users/vehicles (Centralized Trusted Mediator based) | Decentralization | Platform Verification | separation of Duty | Least Privilege Principle | Implementation | DoS/ DDoS attack | Spoofing Attack | Computation cost | Data stored on BC |
|---|---|---|---|---|---|---|---|---|---|---|---|---|
| [9] | Yes | No | No | No | No | Yes | Yes | Single Server | Yes | Yes | High | No Data |
| [40] | No | Yes | No | Yes | No | No | No | Bitcoin | Yes | No | High | Access Policies |
| [41] | No | Yes | No | Yes | No | No | No | Ethereum | No | No | Low | Patient health Records |
| [42] | No | Yes | No | Yes | No | No | No | Ethereum | Yes | No | Low | Access Policies |
| [43] | No | No | Yes | Yes | No | No | No | distributed Software Defined Network based on Network Function Virtualization | No | No | High | User Identities |
| [44] | No | No | Yes | Yes | No | No | No | No Implementation | Yes | Yes | High | No data |
| [45] | No | No | Yes | Yes | No | Yes | Yes | Ethereum | Yes | Yes | Low | Access Policies |
| [46] | No | No | Yes | Yes | No | No | No | No Implementation | Yes | Yes | High | No Data |
| [47] | No | No | Yes | Yes | No | No | No | No Implementation | Yes | Yes | High | No Data |
| [48] | Yes | No | No | No | No | Yes | Yes | Single Server | Yes | Yes | High | No Data |
| [49] | Yes | No | No | No | No | No | No | Single Server | Yes | Yes | High | No Data |
| Proposed Framework | No | No | Yes | Yes | Yes | Yes | Yes | Hyperledger Fabric | No | No | Low | Access Policies & Platform Hashes |

- Storage: We are storing access control policies on BC. However, BC has limited storage capacity, so there exists an opportunity to compact these policies by simplifying the policy structure.

- Performance: The proposed PoIA mechanism shows less computational overhead than its counterparts. However, there is space for improvement to efficiently handle a huge number of users.

## Acknowledgments

This work was supported by the EIAS Data Science and Blockchain Lab, College of Computer and Information Sciences, Prince Sultan University, Riyadh Saudi Arabia.

## Author Contributions

**Conceptualization:** Gauhar Ali.

**Formal analysis:** Gauhar Ali.

**Methodology:** Gauhar Ali, Naveed Ahmad.

**Supervision:** Mohammed ElAffendi.

**Visualization:** Naveed Ahmad.

**Writing – original draft:** Gauhar Ali.

**Writing – review & editing:** Mohammed ElAffendi.

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
