## [Decision Letter · Decision Letter 0]

10 Aug 2023

PONE-D-23-22702BlockAuth: A Blockchain-Based Framework for Secure Vehicle

Authentication and AuthorizationPLOS ONE

Dear Dr. Ali,

Thank you for submitting your manuscript to PLOS ONE. After careful consideration, we feel that it has merit but does not fully meet PLOS ONE’s publication criteria as it currently stands. Therefore, we invite you to submit a revised version of the manuscript that addresses the points raised during the review process.

After the careful evaluation of the manuscript and based on reviewers comments, I suggest the authors to go for a major revision. Authors should carefully address all the reviewer queries and should respond to it.

We look forward to receiving your revised manuscript.

Kind regards,

Shitharth Selvarajan

Academic Editor

PLOS ONE

Journal Requirements:

Reviewers' comments:

Reviewer's Responses to Questions

**Comments to the Author**

1. Is the manuscript technically sound, and do the data support the conclusions?

Reviewer #1: Yes

Reviewer #2: Yes

2. Has the statistical analysis been performed appropriately and rigorously? 

Reviewer #1: Yes

Reviewer #2: No

3. Have the authors made all data underlying the findings in their manuscript fully available?

Reviewer #1: Yes

Reviewer #2: Yes

4. Is the manuscript presented in an intelligible fashion and written in standard English?

Reviewer #1: Yes

Reviewer #2: Yes

5. Review Comments to the Author

Reviewer #1: [1] The abstract should end with a brief statement regarding the significance and impact of this paper.

[2] The method of comparison experiment used in the manuscript is too old. I have not found the recent works published in the year 2022-2023. The following manuscripts can be added.

Shitharth, S.; Yonbawi, S.; Manoharan, H.; Shankar, A.; Maple, C.; Alahmari, S. Secured data transmissions in corporeal unmanned device to device using machine learning algorithm. Phys. Commun. 2023, 59, 102116.

[3] Abstract should be shortened appropriately. Please reduce the content of the abstract and highlight the merits of the proposed scheme.

[4] The manuscript lacks evaluation indicators, please add several evaluation indicators for further comparative analysis.

[5] The conclusions should explain the comparative results between the proposed and state-of-the-art methods.

[6] The work is well written and well presented but needs to be proofread in English as it has some typos.

[7] Although the authors provide a contextualization of the problem in the introduction, it is not clear what the contribution of the article is.

[8] The authors provided a good description of the works in the literature, but did not provide a detailed description of the difference between the proposed work and other works in the literature. I suggest putting in a table the main characteristics found in the literature and also the proposed work, thus demonstrating the difference between the works.

Reviewer #2: The authors have done a good job on the Blockchain-Based Framework for Secure Vehicle, but this paper still needs improvement.

1. The abstract should always mention the rate of efficacy/efficiency percentage of the proposed method for the reader’s quick overview.

2. The abstract should at least have a line or two about the need for this work. This abstract has an intro and it straightaway deals with the proposed work.

3. The novelty of this paper is not clear. The difference between the present work and previous works should be highlighted. Add more of the issues and what is the significance of this research.

4. The paper lacks a convincing theoretical framework, which is necessary to be considered for publication.

5. Nomenclature should be included.

6. The use of more parametric comparisons by the authors is recommended. Confusion matrices are a key component for any system’s validity. However, they are seldom mentioned by the authors.

7. Another major issue of this paper is Missing implementation details. Where is the sample solidity code of that (Hyperledger code)?

8. Among 42 references, hardly three are from 2023. This shows that the paper hasn’t considered many contemporary related works in the survey. I suggest a few more papers to cite and refer.

• https://doi.org/10.4018/978-1-6684-7455-6.ch019

• doi: 10.1038/s41598-023-34354-x

• https://doi.org/10.7717/peerj-cs.1308

• “ An artificial intelligence lightweight blockchain security model for security and privacy in IIoT systems”. J Cloud Comp,(2023), pp.12-38.

9. Unfortunately, the language and sentence structures of this manuscript are at times incomprehensible. The paper needs rewriting and thorough language editing to allow for a proper peer review.

10. Authors are advised to follow the IMRAD format for the entire paper.

11. A separate section for Limitations and future work in detail would give further ideas for the readers who wish to enhance your work.

6. PLOS authors have the option to publish the peer review history of their article (what does this mean?). If published, this will include your full peer review and any attached files.

Reviewer #1: No

Reviewer #2: No

---

## [Author Response · Author response to Decision Letter 0]

29 Aug 2023

Original Manuscript ID: PONE-D-23-22702 

Original Article Title: “BlockAuth: A Blockchain-Based Framework for Secure Vehicle

Authentication and Authorization”

To: PloS ONE Editor

Re: Response to reviewers

Dear Editor,

Thank you for allowing a resubmission of our manuscript, with an opportunity to address the reviewers’ comments.

We are uploading (a) our point-by-point response to the comments (below) (response to reviewers), (b) an updated manuscript with yellow highlighting indicating changes, and (c) a clean updated manuscript without highlights (PDF main document).

Best regards,

Gauhar Ali, et al.

Editor, Concern # 1: Please ensure that your manuscript meets PLOS ONE's style requirements, including those for file naming. The PLOS ONE style templates can be found at 

Author response: Thank you for your valuable suggestion. The manuscript is written using PLOS ONE overleaf format. Both links are visited and the manuscript is formatted according to the given instruction

Author action: We updated the manuscript according to the guidelines given in the files.

Editor, Concern # 2: Please note that PLOS ONE has specific guidelines on code sharing for submissions in which author-generated code underpins the findings in the manuscript. In these cases, all author-generated code must be made available without restrictions upon publication of the work. Please review our guidelines at https://journals.plos.org/plosone/s/materials-and-software-sharing#loc-sharing-code and ensure that your code is shared in a way that follows best practices and facilitates reproducibility and reuse.

Author response: Thank you for your valuable comment. The implementation codes are placed on GitHub. A link to the implementation code is given in the “Data Availability” Section. The following is a GitHub link for the implementation code.

https://github.com/gali675/BlockAuth.git

Author action: We updated the manuscript by adding the “Data Availability” section on page no.20.________________________________________

Editor, Concern # 3: In your Data Availability statement, you have not specified where the minimal data set underlying the results described in your manuscript can be found. PLOS defines a study's minimal data set as the underlying data used to reach the conclusions drawn in the manuscript and any additional data required to replicate the reported study findings in their entirety. All PLOS journals require that the minimal data set be made fully available. For more information about our data policy, please see http://journals.plos.org/plosone/s/data-availability.

Author response: Thank you for your valuable comment. The sample policy data and implementation codes are placed on GitHub. A link to the sample policy data and implementation code is given in the “Data Availability” Section. The following is a GitHub link for the implementation code.

https://github.com/gali675/BlockAuth.git

Author action: We updated the manuscript by adding the “Data Availability” section on page no.20.________________________________________

Reviewer#1, Concern # 1: The abstract should end with a brief statement regarding the significance and impact of this paper.

Author response: Thank you for your valuable comment. The following statement has been added to the abstract.

Furthermore, it opens up global access, enforces the principles of separation of duty and least privilege, and reinforces resilience via decentralization and automation.

Author action: We updated the manuscript by adding the above discussion on page no.1, section “Abstract”.

Reviewer#1, Concern # 2: The method of comparison experiment used in the manuscript is too old. I have not found the recent works published in the year 2022-2023. The following manuscripts can be added.

Shitharth, S.; Yonbawi, S.; Manoharan, H.; Shankar, A.; Maple, C.; Alahmari, S. Secured data transmissions in corporeal unmanned device to device using machine learning algorithm. Phys. Commun. 2023, 59, 102116.

Author response: Thank you for your valuable suggestion. A number of recent works published in the year 2022-2023 are added to the literature review. Also, the above-suggested manuscript is cited in the paper. Additionally, the most relevant manuscripts are added to the “Comparisons among Proposed and Exiting Frameworks” table. The references added to the manuscripts are given below.

[22] Aluvalu R, VN SK, Thirumalaisamy M, Basheer S, Selvarajan S, et al. Efficient data

transmission on wireless communication through a privacy-enhanced blockchain process.

PeerJ Computer Science. 2023;9:e1308.

[23] Saeed H, Malik H, Bashir U, Ahmad A, Riaz S, Ilyas M, et al. Blockchain technology in

healthcare: A systematic review. Plos one. 2022;17(4):e0266462.

[24] Shitharth S, Yonbawi S, Manoharan H, Shankar A, Maple C, Alahmari S. Secured data

transmissions in corporeal unmanned device to device using machine learning algorithm.

Physical Communication. 2023; p. 102116.

[25] Wamba SF, Queiroz MM. Blockchain in the operations and supply chain management:

Benefits, challenges and future research opportunities; 2020.

[26] Manoharan H, Manoharan A, Selvarajan S, Venkatachalam K. Implementation of

Internet of Things With Blockchain Using Machine Learning Algorithm: Enhancement

of Security With Blockchain. IGI Global; 2023.

[27] Selvarajan S, Srivastava G, Khadidos AO, Khadidos AO, Baza M, Alshehri A, et al. An artificial intelligence lightweight blockchain security model for security and privacy in IIoT systems. Journal of Cloud Computing. 2023;12(1):38.

[41] Shitharth S, Mouratidis H. A quantum trust and consultative transaction-based blockchain cybersecurity model for healthcare systems. Scientific Reports. 2023;13(1):7107.

Author action: We updated the manuscript by adding the above references in the “related works” Section on pages no. 3 and 5. Also, it is added to Table 6 “Comparisons among Proposed and Exiting Frameworks”, page no 26.________________________________________

Reviewer#1, Concern # 3: Abstract should be shortened appropriately. Please reduce the content of the abstract and highlight the merits of the proposed scheme 

Author response: Thank you for your valuable comment. The new abstract is given below.

Intelligent Transport System (ITS) offers inter-vehicle communication, safe driving, road condition updates, and intelligent traffic management. This research intends to propose a novel decentralized "BlockAuth" architecture for vehicles, authentication, and authorization, traveling across the border. It is required because the existing architects rely on a single Trusted Authority (TA) for issuing certifications, which can jeopardize privacy and system integrity. 

Similarly, the centralized TA, if failed, can cause the whole system to collapse. Furthermore, a unique "Proof of Authenticity and Integrity" process is proposed, redirecting drivers/vehicles to their home country for authentication, ensuring the security of their credentials. Implemented with Hyperledger Fabric, BlockAuth ensures secure vehicle authentication and authorization with minimal computational overhead, under 2%. Furthermore, it opens up global access, enforces the principles of separation of duty and least privilege, and reinforces resilience via decentralization and automation.

Author action: We updated the manuscript by reducing the content of the abstract (from 16 lines to 11 lines). Also, the merits of the proposed scheme are highlighted on page no 1, section abstract.

Reviewer#1, Concern # 4: The manuscript lacks evaluation indicators, please add several evaluation indicators for further comparative analysis.

Author response: Thank you for your valuable comment. The following evaluation indicators are used in the comparative analysis. Four new evaluation parameters i.e., separation of duty, least privilege principle, DoS/DDOS attack, and spoofing attacks are added to the list of evaluation indicators. Similarly, 

Additionally, the “Threat Models” subsection is added to the main section “ Thread Analysis and Threat Models” with the following discussions.

3.4.2 Threat Models

The following are the potential threats to the proposed BlockAuth framework.

• Malicious User/Vehicle: Suppose a scenario where a malicious user/vehicle wants to get authentication by spoofing a legitimate user/vehicle ID. In the proposed framework "T.join" request consists of the user/vehicle platform hash and pseudonymous ID. The smart contract verifies the user’s platform hash from his country's RCA. The RCA binds user/vehicle pseudonymous IDs with the user’s platform hash during initial registration. As a result, it thwarts any attempt to initiate a spoofing attack. 

• Malicious Service Provider: Suppose a scenario where a malicious SP intends to uncover the true identity of a user/vehicle. In the proposed framework users/vehicles provide their platform hashes and pseudonymous IDs to the SP for the purpose of authentication. The SP subsequently redirects the users/vehicles to their respective country's RCA for authentication, because it stores the real identity of the user/vehicle. As a result, the country RCA provides the stored platform hash of the user/vehicle. Therefore, it is impossible for SP to extract the genuine identity of the user/vehicle from the platform hash.

• DoS/DDoS Attack on Service Provider Authentication Service: Suppose a scenario where an attacker uses a compromised user/vehicle platform and initiates a DoS/DDoS attack to overwhelm the services of the SP. However, the compromised user/vehicle's current platform hash must be different from the hash stored with the user/vehicle's country RCA. This disparity allows SP to deny the malicious access request during the authentication process. As a result, this countermeasure prevents the attacker from initialing a DoS/DDoS attack.

Author action: We updated the manuscript by adding Four new evaluation parameters i.e., separation of duty, least privilege principle, DoS/DDOS attack, and spoofing attacks are added to section 3.4 “comparative analysis”, Table 6, on pages no. 19 and 26. Moreover, section 3.3.2 “Theat Models” is added on page no 19.

Reviewer#1, Concern # 5: The conclusions should explain the comparative results between the proposed and state-of-the-art methods.

Author response: Thank you for the kind comment. The conclusion section is updated. The updated conclusion is given below. 

We proposed a BC-based cross-border vehicle authentication and authorization architecture. The proposed BlockAuth architecture consists of a BC network of CA nodes. The BC network authenticates and authorizes cross-border vehicles. The substitution of the centralized access control service outlined in the existing literature with BC technology mitigates the vulnerability that could result in system failure and various cyber-attacks like DoS and DDOS. Similarly, storing access control policies on the BC ensured the prevention of illegal authorization, thus enhancing the security and integrity of the framework. Unlike existing frameworks, the authorization service is enforced with separation of duty and least privilege principles to ensure that users are not granted more privileges than their necessary needs. Moreover, unlike conventional frameworks, BlockAuth allows user/vehicle authentication from his home country CA. Thus, ensures the protection of user/vehicle credentials. Furthermore, Hyperledger is used to implement the BlockAuth architecture. The BlockAuth ensures secure vehicle authentication and authorization with high throughput and minimal computational overhead, under 2 %. 

Author action: We updated the manuscript by modifying the conclusion section on page no. 20.

Reviewer#1, Concern # 6: The work is well written and well presented but needs to be proofread in English as it has some typos.

Author response: Thank you for the valuable suggestions. The proofread is performed and all typos are corrected.

Author action: We updated the manuscript by thoroughly performing proofreading and correcting all typos.

Reviewer#1, Concern # 7: Although the authors provide a contextualization of the problem in the introduction, it is not clear what the contribution of the article is.

Author response: Thank you for the valuable comment. The “Challenges”, “Our Contributions”, and “Significance of the Study” sections are added to the research article. These sections are given below.

0.1 Challenges

The following are the problems within the existing literature.

• A centralized trusted access control service is a single point of failure and has low resilience to different attacks.

• As a result of inadequate implementation of the separation of duty and least privilege principle, the users are granted excessive privileges. 

• This trusted third-party service can perform illegal authorization by altering stored authorization policies

• This trusted service can expose user credentials without user consent.

0.2 Our contributions

The following are the main contributions of this research study.

• The single trusted authentication and authorization service is substituted with BC because it can collapse the entire system and expose it to several attacks.

• The authorization procedure is strengthened with the implementation of the separation of duty and least privilege principle in the smart contract.

• Our proposed BlockAuth architecture stores authorization policies on the immutable ledger of the BC, effectively preventing any illegal authorization.

• The proposed BlockAuth architecture allows drivers/vehicles to get authentication from their parent countries, thus preventing credentials disclosure.

0.3 Significance of the Study

 Collectively, the highlighted contributions hold immense practical implications. The replacement of the centralized access control service with BC technology mitigates the vulnerability that could potentially lead to the collapse of the entire system and susceptibility to various attacks. Similarly, storing authorization policies on the BC not only ensures that illegal authorizations are effectively prevented but also enhances the security and integrity of the system. Additionally, enabling authentication from parent countries ensures the privacy of credentials, effectively averting the risks associated with credentials disclosure. Thus, this research significantly advances the realm of authentication and authorization, offering enhanced security and efficiency for modern systems.

Author action: We updated the manuscript by adding the “Challenges”, “Our Contributions”, and “Significance of the Study” sections in the “Introduction” section on pages no. 2 and 3.________________________________________

Reviewer#1, Concern # 8: The authors provided a good description of the works in the literature, but did not provide a detailed description of the difference between the proposed work and other works in the literature. I suggest putting in a table the main characteristics found in the literature and also the proposed work, thus demonstrating the difference between the works.

Author response: Thank you for the valuable comment. The main characteristics of the related literature discussed in the related works section are tabulated and compared with a proposed framework in Table 6, titled “Comparisons among Existing and BlockAuth architectures”.

Author action: We updated the manuscript by adding a table of comparisons between existing and proposed frameworks in Table 6, Section III “Comparative Analysis” on page no. 26.

Reviewer#2, Concern # 1: The abstract should always mention the rate of efficacy/efficiency percentage of the proposed method for the reader’s quick overview.

Author response: Thank you for your valuable comment. The efficacy and merits of the proposed scheme are highlighted. The updated abstract is given below.

Intelligent Transport System (ITS) offers inter-vehicle communication, safe driving, road condition updates, and intelligent traffic management. This research intends to propose a novel decentralized "BlockAuth" architecture for vehicles, authentication, and authorization, traveling across the border. It is required because the existing architects rely on a single Trusted Authority (TA) for issuing certifications, which can jeopardize privacy and system integrity. Similarly, the centralized TA, if failed, can cause the whole system to collapse. Furthermore, a unique "Proof of Authenticity and Integrity" process is proposed, redirecting drivers/vehicles to their home country for authentication, ensuring the security of their credentials. Implemented with Hyperledger Fabric, BlockAuth ensures secure vehicle authentication and authorization with minimal computational overhead, under 2%. Furthermore, it opens up global access, enforces the principles of separation of duty and least privilege, and reinforces resilience via decentralization and automation.

Author action: We updated the manuscript by updating the abstract on page no 1, section “Abstract”.

Reviewer#2, Concern # 2: The abstract should at least have a line or two about the need for this work. This abstract has an intro and it straightaway deals with the proposed work.

Author response: Thank you for your valuable comment. The following statement has been added to the abstract.

This research intends to propose a novel decentralized "BlockAuth" architecture for vehicles, authentication, and authorization, traveling across the border. It is required because the existing architects rely on a single Trusted Authority (TA) for issuing certifications, which can jeopardize privacy and system integrity. Similarly, the centralized TA, if failed, can cause the whole system to collapse.

Author action: We updated the manuscript by adding the above discussion on page no.1, section “abstract”.

Reviewer#2, Concern # 3: The novelty of this paper is not clear. The difference between the present work and previous works should be highlighted. Add more of the issues and what is the significance of this research.

Author response: Thank you for the valuable comment. The “Challenges”, “Our Contributions” and “Significance of the Study” sections are added to the research article. Additionally, the present work is compared with previous works and is presented in section III. These sections are given below.

0.1 Challenges

The following are the problems within the existing literature.

• A centralized trusted access control service is a single point of failure and has low resilience to different attacks.

• As a result of inadequate implementation of the separation of duty and least privilege principle, the users are granted excessive privileges. 

• This trusted third-party service can perform illegal authorization by altering stored authorization policies

• This trusted service can expose user credentials without user consent.

0.2 Our contributions

The following are the main contributions of this research study.

• The single trusted authentication and authorization service is substituted with BC because it can collapse the entire system and expose it to several attacks.

• The authorization procedure is strengthened with the implementation of the separation of duty and least privilege principle in the smart contract.

• Our proposed BlockAuth architecture stores authorization policies on the immutable ledger of the BC, effectively preventing any illegal authorization.

• The proposed BlockAuth architecture allows drivers/vehicles to get authentication from their parent countries, thus preventing credentials disclosure.

0.3 Significance of the Study

 Collectively, the highlighted contributions hold immense practical implications. The replacement of the centralized access control service with BC technology mitigates the vulnerability that could potentially lead to the collapse of the entire system and susceptibility to various attacks. Similarly, storing authorization policies on the BC not only ensures that illegal authorizations are effectively prevented, enhancing the security and integrity of the system. Additionally, enabling authentication from parent countries ensures the privacy of credentials, effectively averting the risks associated with credentials disclosure. Thus, this research significantly advances the realm of authentication and authorization, offering enhanced security and efficiency for modern systems. control policies. Moreover, BC makes policy publication and enforcement obvious to all users.

Additionally, the main characteristics of the related literature discussed in the related works section are tabulated and compared with a proposed framework in Table 6, titled “Comparisons among Existing and BlockAuth architectures” 

Author action: We updated the manuscript by adding the “Challenges”, “Our Contributions” and “Significance of the Study” sections in the “Introduction” section on pages no. 2 and 3. Additionally, a comparison between existing and proposed frameworks is added in Table 6, Section III, page no 26. 

Reviewer#2, Concern # 4: The paper lacks a convincing theoretical framework, which is necessary to be considered for publication.

Author response: Thank you for the valuable comments. The theoretical framework is updated by modifying the “vehicle authentication and authorization” subsection to include the separation of duty and least privilege principles. Similarly, four axioms are added to clarify the working of the proposed framework. These axioms are given below.

Axiom 1:

Every vehicle possesses a distinct platform hash value, ensuring that no two vehicles share the same value.

Axiom 2:

If an attacker manages to infiltrate a vehicle's on-board units and installs malicious software, the resulting platform hash value will differ from the previously stored value. Consequently, the vehicle would be unable to successfully authenticate.

Axiom 3:

In the case of multiple policies for a vehicle, the policy with the least privilege will take precedence.

Axiom 4:

This axiom establishes the concept of separation of duty. When two policies present conflicting interests, the vehicle will grant authorization based on one of the policies.

Similarly, Algorithm 1 (smart contract operations) is updated according to the axioms given above.

Author action: We updated the manuscript by modifying subsection 2.2.5” vehicle authentication and authorization” on page no. 10. Moreover, four axioms are added to the 2.3.2 subsection on pages no. 12 and 13. Similarly, Algorithm 1 on subsection 2.3.3 on page no. 14. 

Reviewer#2, Concern # 5: Nomenclature should be included.

Author response: Thank you for the valuable suggestions. Nomenclature is added in Appendix A.

Author action: We updated the manuscript by adding a nomenclature in Appendix A on pages no. 24 and 25.

Reviewer#2, Concern # 6: The use of more parametric comparisons by the authors is recommended. Confusion matrices are a key component for any system’s validity. However, they are seldom mentioned by the authors.

Author response: Thank you for your valuable comment. A confusion matrix is primarily used in the context of classification tasks where you have actual class labels and predicted class labels, allowing you to evaluate the performance of a classification model by comparing how well the predicted labels match the actual labels. If you don't have predictions, the confusion matrix might not provide meaningful insights. 

The confusion matrices cannot be applied to the proposed system because it does not propose a prediction model. However, we evaluate the performance of the proposed BlockAuth using chain size analysis, BC resources usage analysis, throughput analysis, and overhead ratio analysis. 

The following evaluation indicators are used in the comparative analysis. Four new evaluation parameters i.e., separation of duty, least privilege principle, DoS/DDOS attack, and spoofing attacks are added to the list of evaluation indicators. Similarly, 

Additionally, the “Threat Models” subsection is added to the main section “ Thread Analysis and Threat Models” with the following discussions.

3.3.2 Threat Models

The following are the potential threats to the proposed BlockAuth framework.

• Malicious User/Vehicle: Suppose a scenario where a malicious user/vehicle wants to get authentication by spoofing a legitimate user/vehicle ID. In the proposed framework "T.join" request consists of the user/vehicle platform hash and pseudonymous ID. The smart contract verifies the user’s platform hash from his country's RCA. The RCA binds user/vehicle pseudonymous IDs with the user’s platform hash during initial registration. As a result, it thwarts any attempt to initiate a spoofing attack. 

• Malicious Service Provider: Suppose a scenario where a malicious SP intends to uncover the true identity of a user/vehicle. In the proposed framework users/vehicles provide their platform hashes and pseudonymous IDs to the SP for the purpose of authentication. The SP subsequently redirects the users/vehicles to their respective country's RCA for authentication, because it stores the real identity of the user/vehicle. As a result, the country RCA provides the stored platform hash of the user/vehicle. Therefore, it is impossible for SP to extract the genuine identity of the user/vehicle from the platform hash.

• DoS/DDoS Attack on Service Provider Authentication Service: Suppose a scenario where an attacker uses a compromised user/vehicle platform and initiates a DoS/DDoS attack to overwhelm the services of the SP. However, the compromised user/vehicle's current platform hash must be different from the hash stored with the user/vehicle's country RCA. This disparity allows SP to deny the malicious access request during the authentication process. As a result, this countermeasure prevents the attacker from initialing a DoS/DDoS attack.

Author action: We updated the manuscript by adding four new evaluation parameters i.e., separation of duty, least privilege principle, DoS/DDOS attack, and spoofing attacks are added to section 3.4 “comparative analysis”, Table 6, on page no. 20 and 26. Moreover, section 3.3.2 “Theat Models” is added on page no 19.________________________________________

Reviewer#2, Concern # 7: Another major issue of this paper is Missing implementation details. Where is the sample solidity code of that (Hyperledger code)?

Author response: Thank you for your valuable comment. Yes. The implementation codes are placed on GitHub. A link to the implementation code is given in the “Data Availability” Section. The following is a GitHub link for the implementation code.

https://github.com/gali675/BlockAuth.git

Author action: We updated the manuscript by adding the “Data Availability” section on page no.21.________________________________________

Reviewer#2, Concern # 8: Among 42 references, hardly three are from 2023. This shows that the paper hasn’t considered many contemporary related works in the survey. I suggest a few more papers to cite and refer.

• https://doi.org/10.4018/978-1-6684-7455-6.ch019

• doi: 10.1038/s41598-023-34354-x

• https://doi.org/10.7717/peerj-cs.1308

• “ An artificial intelligence lightweight blockchain security model for security and privacy in IIoT systems”. J Cloud Comp,(2023), pp.12-38.

Author response: Thank you for your valuable suggestion. A number of recent works published in the year 2022-2023 are added to the literature review. Also, the above-suggested manuscripts are cited in the paper. Additionally, the most relevant manuscripts are added to the “Comparisons among Proposed and Exiting Frameworks” table. The references added to the manuscripts are given below.

[22] Aluvalu R, VN SK, Thirumalaisamy M, Basheer S, Selvarajan S, et al. Efficient data

transmission on wireless communication through a privacy-enhanced blockchain process.

PeerJ Computer Science. 2023;9:e1308.

[23] Saeed H, Malik H, Bashir U, Ahmad A, Riaz S, Ilyas M, et al. Blockchain technology in

healthcare: A systematic review. Plos one. 2022;17(4):e0266462.

[24] Shitharth S, Yonbawi S, Manoharan H, Shankar A, Maple C, Alahmari S. Secured data

transmissions in corporeal unmanned device to device using machine learning algorithm.

Physical Communication. 2023; p. 102116.

[25] Wamba SF, Queiroz MM. Blockchain in the operations and supply chain management:

Benefits, challenges and future research opportunities; 2020.

[26] Manoharan H, Manoharan A, Selvarajan S, Venkatachalam K. Implementation of

Internet of Things With Blockchain Using Machine Learning Algorithm: Enhancement

of Security With Blockchain. IGI Global; 2023.

[27] Selvarajan S, Srivastava G, Khadidos AO, Khadidos AO, Baza M, Alshehri A, et al. An artificial intelligence lightweight blockchain security model for security and privacy in IIoT systems. Journal of Cloud Computing. 2023;12(1):38.

[41] Shitharth S, Mouratidis H. A quantum trust and consultative transaction-based blockchain cybersecurity model for healthcare systems. Scientific Reports. 2023;13(1):7107.

Author action: We updated the manuscript by adding the above references in the “related works” Section on pages no. 3 and 5. Also, it is added to Table 6 “Comparisons among Proposed and Exiting Frameworks”, page no 26.________________________________________

Reviewer#2, Concern # 9: Unfortunately, the language and sentence structures of this manuscript are at times incomprehensible. The paper needs rewriting and thorough language editing to allow for a proper peer review.

Author response: Thank you for the valuable suggestions. The paper's language and sentence structures are thoroughly reviewed and corrected.

Author action: We updated the manuscript by correcting all the grammar and sentence structure mistakes. 

Reviewer#2, Concern # 10: Authors are advised to follow the IMRAD format for the entire paper.

Author response: Thank you for the valuable suggestions. The paper organization is changed according to IMRAD format. Now, the paper consists of five sections i.e., “Introduction”, “Preliminaries and Related Works”, “Methods”, “Results and Discussion”, and “Conclusion”.

Author action: We updated the manuscript by merging different sections for example, the “Security Analysis” and “Implementation Details” sections are merged into the “Results and Discussion” section. Similarly, the “Overview of BlockAuth Architecture” and “Formal modeling” sections are merged into the “Methods” section.________________________________________

Reviewer#2, Concern # 11: A separate section for Limitations and future work in detail would give further ideas for the readers who wish to enhance your work.

Author response: Thank you for the valuable comments. Limitations and future work sections are added to the paper. These sections are given below.

4.1 Limitations of the proposed BlockAuth Framework: 

The following are the limitations of the BlockAuth framework.

• Scalability: As more CAs become part of the virtual coalition, the count of BC nodes within the overlay network will grow. This expansion increases computational overhead. As a result, the BlockAuth exhibits limited scalability.

• Storage: The proposed BlockAuth stores access control policies on BC. However, these policies consume a larger amount of storage due to their complex structure. 

4.2 Future Directions

 We outline potential avenues for future research exploration.

• Security: Future research could involve BlockAuth resilience against emerging attacks on BC. Additionally, BC security could be enhanced by using formal modeling and verification techniques.

• Storage: We are storing access control policies on BC. However, BC has limited storage capacity, so there exists an opportunity to compact these policies by simplifying the policy structure.

• Performance: The proposed PoIA mechanism shows less computational overhead than its counterparts. However, there is space for improvement to efficiently handle a huge number of users.

Author action: We updated the manuscript by adding the “Limitations of the proposed BlockAuth Framework” and “Future Directions” Sections on page no.20, subsections 4.1 and 4.2 respectively.

---

## [Editor Report · Decision Letter 1]

4 Sep 2023

BlockAuth: A Blockchain-Based Framework for Secure Vehicle

Authentication and Authorization

PONE-D-23-22702R1

Dear Authors, 

We’re pleased to inform you that your manuscript has been judged scientifically suitable for publication and will be formally accepted for publication once it meets all outstanding technical requirements.

Kind regards,

Shitharth Selvarajan

Academic Editor

PLOS ONE

Additional Editor Comments (optional):

We are happy with your revised version and hence we proceed further with acceptance. 
---

## [Editor Report · Acceptance letter]

11 Sep 2023

PONE-D-23-22702R1 

BlockAuth: A Blockchain-Based Framework for Secure Vehicle
Authentication and Authorization 

Dear Dr. Ali:

I'm pleased to inform you that your manuscript has been deemed suitable for publication in PLOS ONE. Congratulations! Your manuscript is now with our production department. 

Kind regards, 

on behalf of

Dr. Shitharth Selvarajan 

Academic Editor

PLOS ONE